# Semantic similarity across languages reflects neurocognitive dimensions shaped by climate

Ze Fu [1], Yuxi Chu [1], Tangxiaoxue Zhang[2], Yawen Li[2], Xiaosha Wang[1,3,4,5] & Yanchao Bi [1,3,4,5,6]

Human languages differ widely, yet they share systematic regularities in the underlying semantic representations being expressed. How such similarities and differences arise remains unclear, in part because semantic theories often lack a principled link to neurocognitive constraints. Drawing on neurocognitive accounts in which semantic knowledge is grounded in biologically salient information dimensions, we examine how environmental factors shape conceptual representations in language. Here we show that word meanings across languages are organized along shared neurocognitive dimensions, while systematic variation along these dimensions is associated with climate. Using word embeddings from 53 languages and behavioral ratings from speakers of 8 languages, we find converging evidence that climatic variables explain semantic variation beyond commonly considered sociocultural factors. Complementary exploratory brain data further suggest climate-related modulation of activity patterns in the right anterior temporal lobe. Together, these findings indicate that semantic representations in language reflect biologically grounded dimensions that are flexibly shaped by long-term environmental conditions.

There are currently over 7000 spoken and signed languages worldwide, each with distinct sound and visual forms and syntactic rules. Do speakers of different languages simply use different word forms to map onto a common conceptual structure, i.e., do people speaking *rose* and 玫瑰 have the same semantic representation? If not, what is lost in translation? This question is part of the classic debate on language universality versus relativity of whether speaking different languages is associated with different cognitions more broadly[1–3]. It seems trivial that both commonalities and variations exist—most languages have words referring to phenomena such as the *sun* or the color *red*, while the words for the object *rose* entail notions of romance in some languages but not in others. The key question is what principles underlie these commonalities and variations across languages. Answers to this question not only address semantic representation principles but also provide a foundation for better communication across languages and cultures.

Rich commonalities and variations have been documented across multiple disciplines, including anthropology and linguistics. Ethnographic descriptions have identified common concepts across semantic domains such as color[4] and emotion[5,6], while also revealing different words for the same perceptual referents, culturally bound words resistant to translation, and different categorization systems[3,7–9]. Recent advances in computational linguistics allow for cross-linguistic comparisons through analysis of word representations

[1]State Key Laboratory of Cognitive Neuroscience and Learning & IDG/McGovern Institute for Brain Research, Beijing Normal University, Beijing, China. [2]Faculty of Psychology, Beijing Normal University, Beijing, China. [3]School of Psychological and Cognitive Sciences and Beijing Key Laboratory of Behavior and Mental Health, Peking University, Beijing, China. [4]IDG/McGovern Institute for Brain Research, Peking University, Beijing, China. [5]Key Laboratory of Machine Perception (Ministry of Education), Beijing, China. [6]Institute for Artificial Intelligence, Peking University, Beijing, China. ✉e-mail: xiaosha_wang@pku.edu.cn; ybi@pku.edu.cn

derived from extensive language corpora (see review[10]). These approaches assess cross-language semantic alignment (e.g., *beautiful* in English and *bella* in Italian) by comparing their distances to a set of anchor words in their respective high-dimensional representational spaces, allowing for meaningful comparisons across languages. The selection of different anchor word sets (whether domain-specific, proximally neighboring, or the entire word space) and the utilization of diverse data types (including word embeddings and colexification networks) have yielded different results and conclusions accordingly, supporting either word semantics being innate[11] or culturally driven[12,13]. It is difficult to evaluate whether these differences reflect those certain aspects of word meaning (approximated by anchor words) are more universal than others across experiments, given the differences in stimuli and computational methods. To this end, what has been missing is an overarching theoretical framework: What aspects (underlying dimensions) of human semantic mechanisms are hypothesized to be universal? How do variations arise from this universal mechanism?

We propose that the intrinsic way in which the human brain represents semantic knowledge offers a strong candidate framework for understanding both cross-linguistic semantic universality and variation. The biological constraints of the human brain are the result of the biological evolution of *Homo sapiens* and lay the foundation for universality (see similar arguments for color space[4] and emotion space[14]), and such a biological structure would respond to different environmental inputs (e.g., naturally varied across different types of climates), resulting in phenomenal variations. Neurocognitive research reveals a consensus framework that semantic representation in the brain is derived from sensory (and language) experiences in ways that respect the specific information processing architectures of the brain: Brain responses to word meanings are distributed along sensorimotor and related associative cortical networks, respecting domains of evolutionary saliency, with activity strength modulated by the meaning's loading on corresponding attributes/domains[15–18]. Lesions in the brain lead to deficits along the lines of these sensory-motor modalities and domains[19–21]. Some aspects of the neural organization are present early in human infancy[22,23], have been found across diverse cultures and languages (e.g., US, UK, Italy, and China), even in individuals with drastic experiential differences such as complete visual deprivation[24–27]. Several mainstream neuroanatomical semantic models—including GRAPES[28], Embodied Abstraction[29], Hub-and-Spoke Model[30,31], and Neural Dual Coding Theory[26]—converge in their recognition of this foundational biological structure underlying semantic representation, but the commonalities and variations along this framework have not been systematically tested across a large number of languages.

We extend this neurocognitive framework of semantic representations, i.e., representing word semantics along the core neuro-cognitive dimensions, to investigate its effectiveness in explaining semantic commonalities and variations across languages. To operationalize, we gleaned 13 (primitive) dimensions that have extensive evidence for their neural correlates, and constructed word semantic representations composed of these 13 dimensions[32] (see Table S1 for example studies and evidence): sensorimotor dimensions of the human brain system (color, shape, taste, smell, sound, touch, and bodily motor) and core cognitive domains (time, space, number, mental-cognition, emotion, and social). Word representations as 13-dimensional vectors were tested across languages: how similar are the words for *rose* in different languages in terms of their loading (relation) patterns with color (e.g., red, blue), shape (e.g., round, square), emotion (e.g., happy, sad, angry), etc.? This neurocognitive approach also offers methodological advantages for cross-linguistic comparison. By focusing on dimensions with established neural correlates, we create a controlled space where both universal patterns and meaningful variations can be understood against a consistent backbone of biological

universality. Such dimensional structures not only reveal fundamental commonalities in semantic organization across languages due to our shared neural architecture, but also filter out potentially idiosyncratic semantic dimensions to highlight variations specifically along sensorimotor and core-cognitive channels.

The neurocognitive framework makes two key predictions (Fig. 1): First, semantic representation derived from this neurocognitive framework better captures the universal structure of semantic representations compared to other distribution-based models (Fig. 2a) that do not directly incorporate the neural architecture, including distributional semantic models[12], psycholinguistic semantic featural models[33,34], and randomized statistical control models. Second, regarding variations along this dimensional structure, the intrinsic assumption of the framework that semantic representation derives from sensory channels predicts that variability is the natural consequence of variables affecting these channel-mediated experiences, including those associated with both natural and cultural environments[35]. This perspective differs from those of previous studies that predominantly emphasized sociocultural factors, including geographic distance[14], communication pressures[36], linguistic history[37], and cultural proximities[12]. These variables were mainly motivated by the significance of communication, word borrowing, and phylogenetic history in the cultural evolutionary process for cross-language alignment[14,38]. However, beyond such language-related experiences, it is possible that languages used in distant locations share similar conceptual meanings when they have comparable sensory signals and similar neurocognitive structures. One salient ecological variable worth highlighting is climate, which has been shown to be strongly associated with natural and cultural properties[39–41]. By systematically considering these macroscale environmental variables, we predict that climate, as an ecological factor shaping sensory environments, exerts independent effects on human semantic processes beyond previously tested sociocultural-centered variables (geographic, linguistic, and cultural distance).

Here, we demonstrate that the neurocognitive dimensional framework captures cross-language semantic commonalities more effectively than alternative distributional models, and that climate emerges as a significant ecological predictor of semantic variations across languages, independent of sociocultural factors. Converging evidence from a series of studies employing computational, behavioral, and neural approaches supports these conclusions (Fig. 1). Study 1 involved language computational analyses on large-scale multilingual pre-trained word embedding data[42]. For the universality prediction, we compared the extent of commonalities captured by the target model (with the 13 neurocognitive dimensions as anchor words) and alternative models (different types of anchor words). For the variation prediction, we examined the relationship between variations along this neurocognitive dimensional structure and ecological variables across languages. Such models have the advantage of analyzing large-scale language patterns, but the relationship of ecological variables with the human mind/brain is approached indirectly through computational relationships. Thus, in Study 2, we collected human individual semantic behavioral ratings on these 13 dimensions, and in Study 3, we analyzed brain activities during language comprehension, offering preliminary insights into linking cognitive and biological variations more directly with the ecological variables of interest. Figure 1 (bottom panel) illustrates the geographic distribution of language samples across three studies, with detailed language information provided in Table S2.

## Results

### Commonalities on the neurocognitive semantic structure

We examined cross-language semantic commonalities through computational analysis of word embeddings across 53 languages (Study 1a). We focused on the NorthEuraLex (NEL) wordlist, which provides

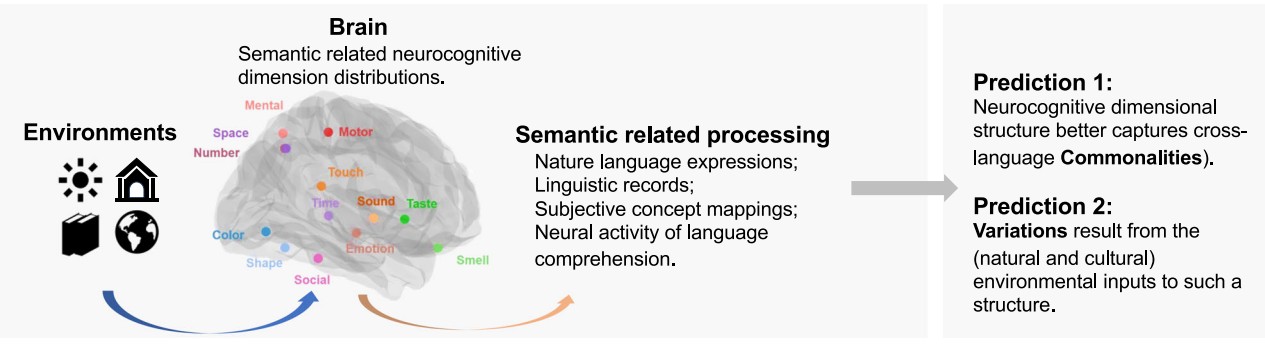

## Theoretical Framework

**Brain**
Semantic related neurocognitive dimension distributions.

**Environments**

**Semantic related processing**
Nature language expressions;
Linguistic records;
Subjective concept mappings;
Neural activity of language comprehension.

**Prediction 1:**
Neurocognitive dimensional structure better captures cross-language **Commonalities**).

**Prediction 2:**
**Variations** result from the (natural and cultural) environmental inputs to such a structure.

## Study Overview

| | | Data | Analysis | Results |
|---|---|---|---|---|
| **Prediction 1** Commonalities | **Study 1a: Language model study** | Pretrained word embedding (N = 53 languages) | Compare representational commonalities of neurocognitive dimensional models against other theoretical and control models. | Neurocognitive model > other controls |
| | | CILCS network (N = 2,681 languages) | Generalize to larger language samples by predicting topological structures of colexification network. | Neurocognitive model > other controls |
| **Prediction 2** Variations | **Study 1b: Language model study** | Pretrained word embedding (N = 53 languages) | Predict semantic variations based on climate, cultural, geographic, and linguistic distances. | Climate Linguistic |
| | **Study 2: Human rating study** | Human participant ratings (N = 8 languages, 253 participants) | Predict semantic variations based on climate, cultural, geographic, and linguistic distances. | Climate |
| | **Study 3: fMRI study** | Multi-language fMRI patterns (N = 45 languages, 86 participants) | Predict neural correlates of language-associated climate and cultural distances. | Climate (r-ATL) |

● Pretrained word embedding **(Study1)**   △ Human participant ratings **(Study2)**   □ fMRI activation patterns **(Study3)**

+ CILCS Data **(Study1: Validation of commonalities)**

Afro-Asiatic   Atlantic-Congo   Austronesian   Austroasiatic
Basque   Dravidian   Indo-European   Japonic
Kartvelian   Koreanic   Sino-Tibetan   Turkic   Uralic

**Geographic distributions**

**Fig. 1 | Theoretical framework and study overview.** We investigated organizing principles of cross-language semantic space by examining commonalities and variations in natural language expressions, linguistic records, subjective concept mappings, and neural activities. Our theoretical framework proposes that (1) a shared neurocognitive dimensional structure (biologically constrained semantic representations) captures cross-language commonalities, while (2) variations arise from natural and cultural environmental inputs. Three studies tested these predictions: (1) Language model study using pretrained word embeddings (53 languages) and cross-linguistic colexification networks (CILCS; 2681 languages); (2) Human rating study (8 languages, 253 participants); (3) fMRI study (45 languages, 86 participants). Upper left: Brain regions showing probabilistic peak activations for semantic processing domains (e.g., social, sensorimotor, emotional), generated via NeuroQuery meta-analysis[75]; r-ATL, right anterior temporal lobe. Bottom: Geographic distribution of language samples by family, with coordinates from Glottolog 4.6[50] or participant self-reports (Study 2).

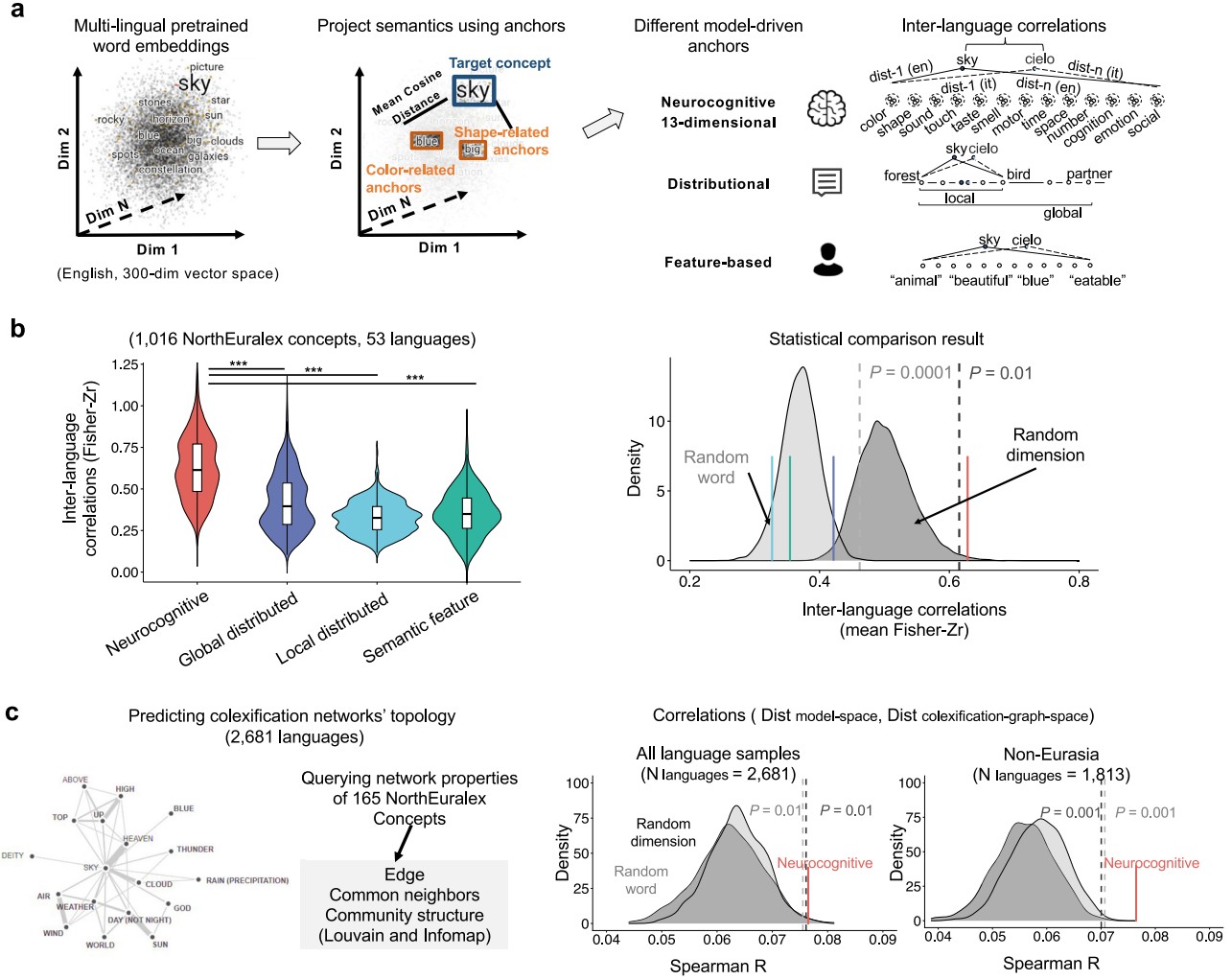

**Fig. 2 | Commonality of the neurocognitive semantic structure across languages using language computational analysis (Study 1a). a** Semantic construction methods in the pre-trained word embedding data analyses. Pre-trained 300-dimensional word embedding data from 53 languages from 10 language families were included. Embedding vectors of target concepts were projected onto three theoretical semantic representations and compared for cross-language commonalities: neurocognitive, distributional (local and global), and feature-based models, each with different anchor words/dimensions (see "Methods"). **b** Semantic commonality results. Left panel: Violin plots show the distribution of the inter-language correlations (Fisher-Z transformed) across 53 languages ($N = 1378$ language pairs). White boxplots inside violins show the median (center line), 25th–75th percentiles (box limits), and whiskers extending to $1.5 \times$ IQR. ***$P < 0.001$; two-sided paired Wilcoxon signed-rank tests with FDR correction for multiple comparisons. Right panel: Comparisons of mean inter-language correlations between the four semantic models and two random control models: random word models (randomly selecting 13 NEL words as anchor words 10,000 times; light gray

area) and random dimension models (randomly selecting 100 NEL words and grouping them into 13 dimensions 10,000 times; dark gray area). The mean inter-language correlation of the neurocognitive model (the red line) was significantly higher than the random word models ($P < 0.001$, one-tailed) and the random dimension models ($P = 0.005$, one-tailed); those of the other models were not. **c** Generalization of 13-dimensional neurocognitive space to larger language samples using the colexification network data. Left panel: Visualization of the CLICS data and network properties representing semantic distance in the colexification topology space. Right panel: Density plots comparing association patterns between semantic distances in the original 53-language embedding-derived neurocognitive space and the colexification topological graph space. The analysis includes all language samples ($N = 2681$) and non-Eurasian languages ($N = 1813$; encompassing South America, North America, Africa, Papunesia, and Australia, excluding unclassified languages). The neurocognitive model's performance (red line) is at the upper bound of the random model distributions, demonstrating robust cross-linguistic generalizability.

1016 concepts across 107 languages. Among them, 53 languages (spanning 10 language families) were well covered in the recently developed word embedding databases (fastText multilingual word vectors[42]). These databases provided 300-dimensional (hidden) word embedding vectors for a large number of languages, pre-trained on vast corpora comprising billions of words for each language. For each target concept in each language, we projected its embedding onto different sets of anchor words based on different models (see below), i.e., computing the target-anchor cosine distances (see an illustration of construction processes in Fig. 2a; see also the anchor word approach[43–45]). Distances to different model-driven anchor sets were taken as the model-based semantic representations for the target

concepts, based on which cross-language comparisons were performed.

We performed cross-linguistic comparisons of semantic representations obtained based on different anchor-based semantic models from word embedding data. All semantic representations were obtained using an anchor word approach (i.e., essentially all are distributional semantic methods), and the key difference is what types of words were used as anchors for representations. Different anchor words were selected based on different model principles:

Distributional models represent concepts by their usage contexts, a prevalent approach in cross-linguistic semantic comparisons[12]. Following this approach, we represented the 1016 concepts using their

local patterns (i.e., anchor words were semantic neighbors, $N = 100$) and global patterns (i.e., anchor words were the entire native word space).

Semantic feature models in the psycholinguistic tradition are based on human-generated decompositional features (e.g., has fur, is round), representing concepts as binary or weighted vectors indicating the presence/absence or strength of each feature[33,34]. To adapt this approach for cross-linguistic comparison, we represented the 1016 concepts using their embedding distance patterns with the 100 most frequent feature words from Buchanan et al.[33].

Neurocognitive semantic models, like semantic feature models, assume decompositional representation but highlight a set of underlying primitive dimensions constrained by the human brain. We identified 13 primary structural dimensions from neurocognitive semantic research (see Introduction and Table S1 for a list of example empirical references). Anchor words corresponding to these dimensions were manually selected from the NEL concept list (see "Methods" for details; anchor word list shown in Table S3). Concepts were projected onto these 13 neurocognitive salient dimensions by averaging semantic distances of anchor words within each dimension. Our text-derived dimension loadings significantly correlated with the human rating data reported in Binder et al.[32]. (Pearson $r_{(6914)} = 0.42$, $P < 0.001$, 95% confidence interval (CI) [0.40, 0.44]; see Fig. S1 for details), consistent with previous findings on subjective-rating validity for the dimension computation[45].

To assess the explanatory powers of the above candidate theoretical models for cross-linguistic semantic universality, we further computed two types of theory-free benchmark distributions. One type is random word models, where 100 randomly selected words from the original NEL list served as anchors, and concepts were projected onto these anchors' embedding vectors, with 10,000 iterations. The other is the random dimension model (dimensionality-matched control), where 100 randomly selected words were grouped into 13-dimensional anchors using K-means clustering, and concepts were then projected onto these pseudo dimensions, with 10,000 iterations. The statistical confidence for each theoretical model's universality was established by comparing its position with these two models' distributions.

To quantify and compare the universality of these semantic models (both candidate theoretical models and random control models), we used two methods for convergence: inter-language correlation analysis (Fig. 2b) and principal component analysis (PCA; Supplementary Note 1 and Fig. S2).

For inter-language correlation analysis, we computed Fisher z-transformed Pearson correlation coefficients between model-based semantic representations for each language pair. As shown in Fig. 2b (left panel), the neurocognitive-dimensional semantic representation had greater cross-language similarity compared to the other representation types (mean Fisher z-transformed correlation: neurocognitive ($M = 0.63$, $SD = 0.19$); distributional (global) ($M = 0.42$, $SD = 0.17$); distributional (local) ($M = 0.33$, $SD = 0.10$); semantic feature ($M = 0.35$, $SD = 0.13$). Two-sided paired Wilcoxon signed-rank tests with false discovery rate (FDR)-correction for 3 comparisons revealed all adjusted $P$s < 0.001: neurocognitive vs. distributional (global), $V = 950123$, $n = 1378$, median difference = 0.21, 95% CI [0.20, 0.21]; neurocognitive vs. distributional (local), $V = 950131$, $n = 1378$, median difference = 0.30, 95% CI [0.29, 0.31]; neurocognitive vs. semantic feature, $V = 750925$, $n = 1225$, median difference = 0.30, 95% CI [0.30, 0.31]). These results were robust when we varied the number of anchor words in the distributional (local) and feature models (see Fig. S3). Figure 2b (right panel) illustrates that the mean inter-language correlation of neurocognitive models (red vertical line) exceeded the upper ends of distributions of random word models (light gray area; $P < 0.001$, one-tailed) and the dimensionality-matched random dimension models (dark gray area; $P = 0.005$, one-tailed), whereas the other types of theoretical models did not demonstrate this pattern. These results

demonstrated that the model's performance stems from the specific neurocognitive dimension rather than anchor word selection and dimensionality reduction procedure per se.

We also employed a complementary PCA on 204 concepts common to all 53 languages, treating languages as features and semantic patterns as samples. This approach assumes that a universal semantic structure would manifest as a relatively high first component (PC1), with the variance explained by PC1 reflecting the degree of universality[46,47]. The neurocognitive-dimensional representation of concepts again showed a greater amount of universality (neurocognitive: 44.31%; global-distributional: 34.16%; local-distributional: 36.38%; feature-based: 34.45%). Similar to the inter-language correlation findings, in comparison with statistical random models, the neurocognitive model, not the other models, was positioned at the upper bounds of their distributions, signifying a relatively universal semantic structure (random word models: $P < 0.001$, one-tailed; random dimension models: $P = 0.009$, one-tailed). Detailed results, including the scree plot and neurocognitive structure PC1 matrix, are presented in Supplementary Note 1 and Fig. S2. We further validated the main analyses above using text embedding data derived from different corpora (Supplementary Note 2 and Fig. S4).

To test the generalizability of neurocognitive semantic structures beyond these initial 53 languages, we investigated whether the neurocognitive semantic model could predict word relational patterns in a larger, more diverse language sample. Given the challenges of obtaining semantic representations for low-resource languages, we utilized the database of Cross-Linguistic Colexifications (CLICS), which contains colexification patterns from 2681 languages[48]. We employed common topological network metrics, including edge, common neighbors, and two community-based measures (Louvain and Infomap algorithms), and used the average distance across these four metrics as the word distance in the topological graph space. The hypothesis here is that if the neurocognitive semantic model outperforms statistical control models in predicting word topological patterns in the colexification network, it would indicate the generalizability of these structures to a broader range of languages. Specifically, we examined whether concepts closer in the 13-dimensional neurocognitive space (derived from embedding data of the original 53 Eurasian languages) were more likely to be topologically proximate in the colexification network.

We conducted Spearman correlations between 165 NEL concepts in the embedding-constructed neurocognitive semantic space and the CLICS topological graph space (Fig. 2c, left panel). For statistical comparison, we generated random models (word and dimension) and correlated them with the CLICS graph space 1000 times to establish null distributions. These tests were performed on the following samples: (1) the entire language sample ($N = 2681$) and (2) non-Eurasian languages (South America, North America, Africa, Papunesia, and Australia; $N = 1813$, excluding unclassified languages). The second test specifically targeted other language samples that were not included in our initial word embedding analysis.

As shown in Fig. 2c (right panel), the neurocognitive dimensional representation derived from the original 53 languages significantly explained topological similarities in the colexification network for both the 2681 languages (random word: $P = 0.006$, one-tailed; random dimension: $P = 0.009$, one-tailed) and the non-Eurasian subset (both baselines: $P$s < 0.001, one-tailed). In both cases, the neurocognitive model's performance was at the upper bound of the random model distributions, providing positive evidence for the cross-language generalizability of these semantic structures. Separate density plots for each non-Eurasian area's language samples yielded similar patterns to the entire sample (Fig. S5).

In summary, the 13-neurocognitive-dimensional structure, obtained from brain studies in a few industrialized languages (see Table S1), captures cross-language similarities better than control

models in the word embedding spaces of 53 languages, which further generalizes to explaining word similarities across a larger and more diverse set of languages in the CLICS networks (colexification data).

## Variations associated with environmental variables

Having established the superiority of the neurocognitive semantic model in capturing semantic universality, we then investigated variations across languages along this universal structure. Based on the shared assumption of semantic neurocognitive theories that semantic representations are derived from sensorimotor (and language) experiences, we hypothesized that variations in the model stem from variables affecting these channels of experience, including both natural and cultural environmental factors. The following variables were considered: climate, geographic, linguistic history, and cultural variables. The question under scrutiny was whether and how variations across languages along the universal neurocognitive semantic structures could be explained by these external variables (i.e., whether language pairs with similar environmental characteristics exhibit similar semantic representations).

To this end, we implemented a consistent analytical framework across all three studies (i.e., studies with word embeddings, behavioral ratings, and neural activity data). At the language-level (aggregating participants where applicable), we constructed linear mixed models that included all four variables of interest simultaneously—geographic, cultural, linguistic history, and climate variables—with random intercepts for language families to account for phylogenetic relatedness[12]. This approach allowed all factors to compete within the same model, revealing their unique contributions to semantic variations. For Studies 2 and 3, we additionally conducted predictions at the individual participant level, confirming that patterns remained consistent across analytical scales.

## Variations in word embeddings associated with climate

We first applied this framework to computational analysis of word embedding data across 53 languages (Study 1b). We employed representation similarity analysis (RSA)[49] to quantitatively model cross-language variations in semantic representational geometries by regressing the semantic dissimilarity representational matrix (RDM; Fig. 3a) on the RDMs of our selected environmental variables. Data on environmental variables for 53 languages were extracted from various official public databases using latitude and longitude provided by Glottolog 4.6[50]. The sample sizes (i.e., language pairs) varied depending on the availability of the environmental data.

Correlation analyses first revealed moderate associations between these four environmental RDMs and the semantic RDM (Spearman $\rho$s = 0.39–0.53, all $P$s < 0.001; Table S4). Within a sample of 29 languages (406 language pairs) where all four environmental variables were available, they collectively explained 37.99% of the semantic (neurocognitive) space variations (Spearman $\rho$ = 0.52, $P$ < 0.001, 95% CI [0.43, 0.60]; Fig. 3b).

We then employed a linear mixed regression model to assess the unique effect of each environmental factor. As indicated in Fig. 3c, climate showed the strongest unique explanatory effects ($\beta$ = 0.28, $P$ < 0.001, 95% CI [0.19, 0.37]). Linguistic history also showed significant effects ($\beta$ = 0.22, $P$ < 0.001, 95% CI [0.16, 0.28]). Cultural and geographic distance did not show significant unique contribution to semantic variations (culture: $\beta$ = 0.07, $P$ = 0.097, 95% CI [−0.01, 0.15]; geography: $\beta$ = 0.05, $P$ = 0.237, 95% CI [−0.03, 0.13]). Our focus on sensorimotor-related influences led us to further investigate climate effects across semantic dimensions. Further analyses revealed that climate contributed significant effects across 12 of the 13 semantic dimensions (all FDR-corrected $P$s < 0.05), with the exception of the shape dimension (FDR-corrected $P$ = 0.118; Fig. 3c, right panel). The climate effects were robust across multiple validation analyses, including those utilizing alternative pre-trained word embeddings and

different random effect structures controlling for non-independence (see Supplementary Notes 2 and 3 and Fig. S4).

We further tested whether the observed cross-linguistic associations between semantic variation and climate were specific to the neurocognitive semantic structure. Linear mixed regression models were employed with climate and other semantic structures as regressors to predict neurocognitive semantic variations and vice versa. Results in Table 1 indicated that the relationship between climate and neurocognitive semantic structure remained significant even when controlling for distributional and feature-based models. Specifically, climate maintained a significant association with the neurocognitive semantic structure after controlling for the distributional global ($\beta$ = 0.03, $P$ < 0.001, 95% CI [0.02, 0.05]), distributional local ($\beta$ = 0.11, $P$ < 0.001, 95% CI [0.09, 0.13]), and feature norm ($\beta$ = 0.12, $P$ < 0.001, 95% CI [0.10, 0.15]) models. Conversely, when predicting these alternative semantic structures while controlling for the neurocognitive structure, climate showed significant associations with distributional global ($\beta$ = 0.04, $P$ < 0.001, 95% CI [0.02, 0.06]), and not with distributional local ($\beta$ = −0.01, $P$ = 0.271, 95% CI [−0.03, 0.01]) or feature norm ($\beta$ = −0.02, $P$ = 0.220, 95% CI [−0.05, 0.01]). These findings suggest a specific relationship between climate and neurocognitive semantic organization that cannot be fully explained by other semantic representation frameworks.

## Variations in behavioral ratings associated with climate

To investigate whether the principles of commonalities and variations observed in language records (Study 1) are reflected in language speakers' semantic behaviors, we conducted a behavior rating study with participants from 8 languages (Study 2). We recruited participants from 58 city sites representing 8 languages (a subset of 53 languages in Study 1) with broad coverage of geographic, linguistic, and cultural diversity: Arabic (Egypt), Chinese (China), English (USA), Hindi (India), Japanese (Japan), Korean (South Korea), Russian (Russia), and Spanish (Spain). A total of 253 participants from these countries were included in the final sample (sampling procedures and geographical information are provided in "Methods", Fig. S6 and Table S5). Participants were asked to rate the Swadesh concepts ($N$ = 207, Table S6) on their associations with the 13 neurocognitive dimensions, resulting in individual semantic spaces comprising 2691 ratings per participant (207 Swadesh concepts × 13 neurocognitive dimensions). We computed Pearson correlations between the rating vectors (207 concepts × 13 dimensions) for each participant pair, resulting in an intersubject correlational matrix (Fig. 3d). The matrix showed a substantial semantic component shared across participants (65.78% of variance), with much smaller language-specific effects (4.20% of variance), validating the commonality of such neurocognitive semantic structures across languages (see Supplementary Note 4 for details).

We first computed average ratings for each concept-dimension pair within each language to create language-level semantic representations. To assess the relationship between neurocognitive semantic space variation and environmental variables, we collected macroenvironmental variables (identical to those in Study 1) for each language and constructed language-level environmental RDMs for climate, culture, geography, and linguistic history (Fig. 3e). We also included a demographic RDM to account for variations in participants' age, gender, education level, and socioeconomic status across language groups.

Hierarchical regression revealed that, in addition to the explanatory power of demographic distance ($R^2$ = 0.12), the four environmental variables collectively accounted for an incremental 47.01% variance in the neurocognitive semantic structural variations (Spearman $\rho$ = 0.74, $P$ < 0.001, 95% CI [0.50, 0.88]). In alignment with our analytical approach in Study 1, we performed analyses at the language-level, with all four environmental variables as simultaneous predictors

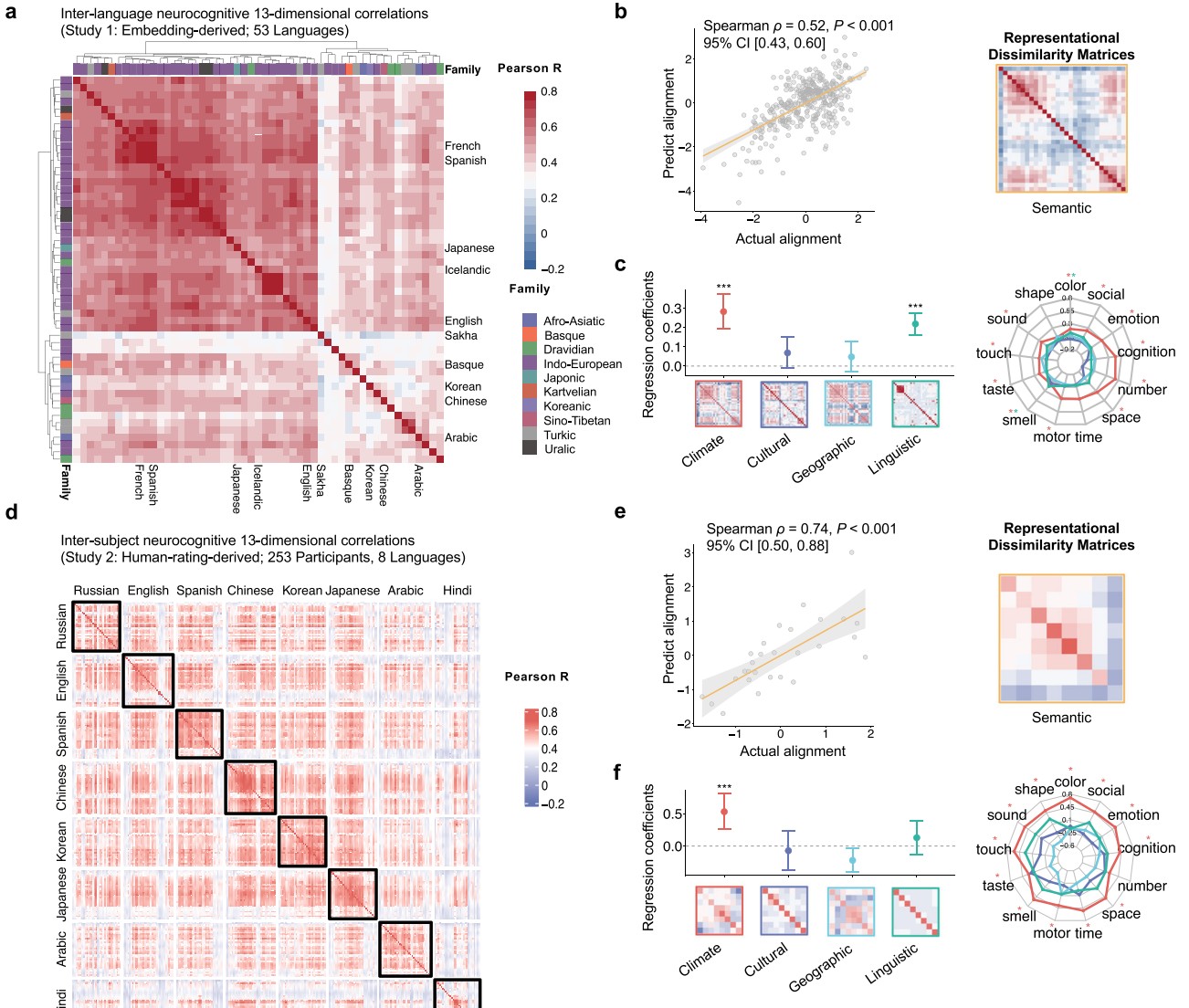

**Fig. 3 | Environmental predictors of neurocognitive semantic variations in word embedding data (Study 1b) and human rating data (Study 2). a** Inter-language correlation matrix of 53 languages based on the 13 neurocognitive dimensional structure. The dendrogram shows the hierarchical clustering of languages by correlation distances, with languages color-coded according to language family. **b** Scatter plot showing the relationship between actual and predicted semantic alignment based on environmental RDMs (climate, linguistic history, geography, and culture) ($N = 29$ languages, 406 unique language pairs). Each point represents a language pair. Shaded area represents 95% CI of the regression line. **c** Linear mixed regression models predicting cross-language semantic variations with environmental RDMs. Left panel: Standardized regression coefficients for each factor are shown as points, with corresponding RDM displayed below. Right panel: Radar plot displaying standardized beta coefficients from regression models for each of the 13 neurocognitive dimensions across environmental variables. **d** Inter-subject correlation matrix of the neurocognitive semantic representations ($N = 253$ participants, 8 languages), with each cell representing the Pearson correlation

coefficient of the ratings on 207 concepts across 13 dimensions (2691 ratings per participant pair). Cells in the black box are the intra-language correlations. **e** Scatter plot showing the relationship between actual and predicted semantic alignment based on environmental variables ($N = 8$ languages, 28 unique language pairs). Each point represents a language pair. Shaded area represents 95% CI of the regression line. **f** Linear mixed regression models predicting the inter-subject language semantic correlation distance using four environmental RDMs (climate, linguistic history, geography, and culture) with a demographic RDM (the Euclidean distance of participants' age, gender, education level, and SES) as the control variable. Left panel: Standardized regression coefficients for each factor are shown as points, with corresponding RDM displayed below. Right panel: Radar plot displaying standardized beta coefficients from regression models for each of the 13 neurocognitive dimensions across environmental variables. Error bars represent 95% CIs. Two-sided tests from linear mixed-effects models: ***, $P < 0.001$; **, $P < 0.01$; *, $P < 0.05$.

and random intercepts for language families to account for phylogenetic relationships. Beta estimation (Fig. 3f) showed that climate had unique effects on cross-language semantic variation (climate: $\beta = 0.53$, $P < 0.001$, 95% CI [0.26, 0.80]). The other three variables showed no or negative effects (culture: $\beta = -0.07$, $P = 0.640$, 95% CI [−0.36, 0.25]; geography: $\beta = -0.22$, $P = 0.025$, 95% CI [−0.40, −0.04]; linguistic history: $\beta = 0.13$, $P = 0.336$, 95% CI [−0.17, 0.39]). When analyzing each neurocognitive dimension separately (Fig. 3f, right panel), all

dimensional variations, except for number, continued to be significantly modulated by climate (number: $q = 0.098$). To examine whether these climate effects extend beyond language-level variation to individual semantic representations, we conducted supplementary analyses while controlling for non-independence (see Supplementary Note 3). These analyses confirmed that climate effects remain robust across different random effect structures and are observable at both language and individual participant levels.

**Table 1 | Unique effects of climate on different semantic structures**

| Model Type | Climate effect (β) | 95% CI | P Value |
|---|---|---|---|
| Model 1: Neurocognitive as DV | | | |
| Neurocognitive ~ Climate + Global Dist. | 0.03 | [0.02, 0.05] | <0.001 |
| Neurocognitive ~ Climate + Local Dist. | 0.11 | [0.09, 0.13] | <0.001 |
| Neurocognitive ~ Climate + Feature | 0.12 | [0.10, 0.15] | <0.001 |
| Model 2: Alternative models as DV | | | |
| Global Dist. ~ Climate + Neurocognitive | 0.04 | [0.02, 0.06] | <0.001 |
| Local Dist. ~ Climate + Neurocognitive | −0.01 | [−0.03, 0.01] | 0.271 |
| Feature ~ Climate + Neurocognitive | −0.02 | [−0.05, 0.01] | 0.220 |

*DV* dependent variable, *Global Dist.* global distributional model, *Local Dist.* local distributional model, *Feature* semantic feature model; Climate effects estimated using linear mixed-effects models with random intercepts for language families. All variables were standardized. *P* values are based on two-sided *t*-tests from linear mixed-effects models.

## Variations in language evoked brain activity patterns associated with climate

To investigate whether the principles of commonalities and variations observed in language records (Study 1) are reflected in language speakers' neural responses during language comprehension, we analyzed a multi-language functional magnetic resonance imaging (fMRI) dataset comprising 86 individuals recruited from the United States, whose native languages spanned 45 languages across 12 language families (Study 3)[51]. We focused on neural activity during native language processing, specifically examining the contrast between intact and acoustically degraded-language conditions in the 12 language-responsive regions (6 left, 6 right) in the language network[52] (Fig. 4a).

Following our consistent analytical approach across studies, we aggregated individual neural responses at the language level. We first investigated whether neural activity patterns in these regions reflected cross-language alignment in the 13-neurocognitive-dimensional semantic space (based on embedding data in Study 1; Fig. 3a). To this end, for overlapping language samples in two studies (65 participants, 33 languages; Fig. 4b), we computed language-level neural RDMs for each brain region by averaging *t*-value maps from speakers of the same language. We used linear mixed regression models to assess the relationships between these neural RDMs and the 13-dimensional neurocognitive semantic model RDMs from Study 1, while including random intercepts for language families to account for the phylogenetic structure. Significant beta coefficients were found only in the right anterior temporal lobe (r-ATL; $\beta = 0.44$, FDR-corrected $P = 0.016$, 95% CI [0.17, 0.71]). That is, the more closely aligned two languages are for the 13-neurocognitive dimensional semantic representation, the more similar their speakers' brain activities are in the r-ATL when processing language. Given the r-ATL's role in semantic processing and the significant relationships with our semantic space representation, we identified this region as our primary region of interest for subsequent analyses (Fig. 4c). We also found that r-ATL showed a certain degree of cross-language commonalities, validating our commonality results (Supplementary Note 4).

To maintain analytical consistency with Studies 1 and 2, we constructed language-level environmental RDMs based on climate, cultural, geographic, and linguistic history distances (Fig. 4c; see "Methods"), analyzing 25 languages for which all environmental measures were available. Linear mixed regression models revealed that climate distance significantly predicted neural pattern dissimilarities in the r-ATL ($\beta = 0.12$, $P = 0.010$, 95% CI [0.03, 0.22]), while the other three distances did not reach significance (all FDR-corrected *P*s > 0.05). This climate effect remained significant ($\beta = 0.13$, $P = 0.007$, 95% CI [0.03, 0.22]) even when controlling for differences in peak activation location (calculated as Euclidean distances between individual peak cluster centroids), indicating that the effect reflects distributed activation patterns across r-ATL voxels rather than simple spatial displacement. To assess robustness, we conducted additional analyses with alternative random effect structures (see Supplementary Note 3), which consistently showed that climate effects remain significant across different model specifications.

Importantly, when using environmental RDMs to predict the r-ATL's neural activity during non-linguistic tasks (e.g., Spatial Working Memory–Hard vs Easy; Math–Hard vs Easy), the climate effects were no longer significant, suggesting that these effects are unique to language-related processing (Supplementary Note 5 and Fig. S7). To further characterize the relationship between semantic and climate effects in the r-ATL, we conducted a commonality analysis to partition the r-ATL variance explained by different components (Fig. 4c, right panel). This analysis revealed unique contributions from both semantic factors (6.26%, 95% CI [1.00%, 37.46%]) and climate factors (55.58%, 95% CI [16.83%, 85.80%]), as well as substantial common variance shared between climate and semantic effects (38.16%, 95% CI [14.75%, 48.30%]). The common variance suggests that climate effects on neural activities in the r-ATL partially reflect semantically related neural activation patterns.

In summary, cross-language variations on the 13-neurocognitive-dimension structure obtained from language computation data (Study 1), human behavior rating data (Study 2), and multi-language fMRI data (Study 3), were all significantly predicted by climate, the effects of which were also robust across all validation analyses (see Table 2). Our neural findings in Study 3 revealed that climate-related semantic variations were specifically associated with activity patterns in the r-ATL, a critical region for semantic processing. Given the limited sample size per language, these findings warrant replication with larger cohorts.

## Exploratory analyses on the associative patterns between climate and semantic space

Having identified that climate has robust and unique effects on variations in semantic structures across large text models (Study 1b), subjective ratings (Study 2), and brain activity patterns (Study 3), we aimed to elucidate the semantic profiles that are associated with the major climate groups. We carried out the following analyses based on the embedding data from Study 1 given that this study covers the largest language sample and concept set. We performed a PCA on climate data consisting of 19 biologically relevant climate variables related to temperature and precipitation across 53 languages. The results revealed two primary climate-related principal components (PCs): Climate PC1, accounting for 42.30% of the variation, and Climate PC2, accounting for 30% (Fig. 5a). The contributions of specific climate variables to these PCs (Table S7) led us to characterize the Climate PC1 as representing cold/temperate vs. tropical climates and the Climate PC2 as representing oceanic vs. continental climates (more precipitation and low seasonality vs. less precipitation and high seasonality).

To project the semantic space along each PC axis, we scaled the semantic space for each language and multiplied it by their loadings on the two climate PCs, resulting in semantic spaces along the Climate PC1 and Climate PC2 axes (Fig. 5b). Higher values along a particular direction of PCs indicate that the given climate type tends to have

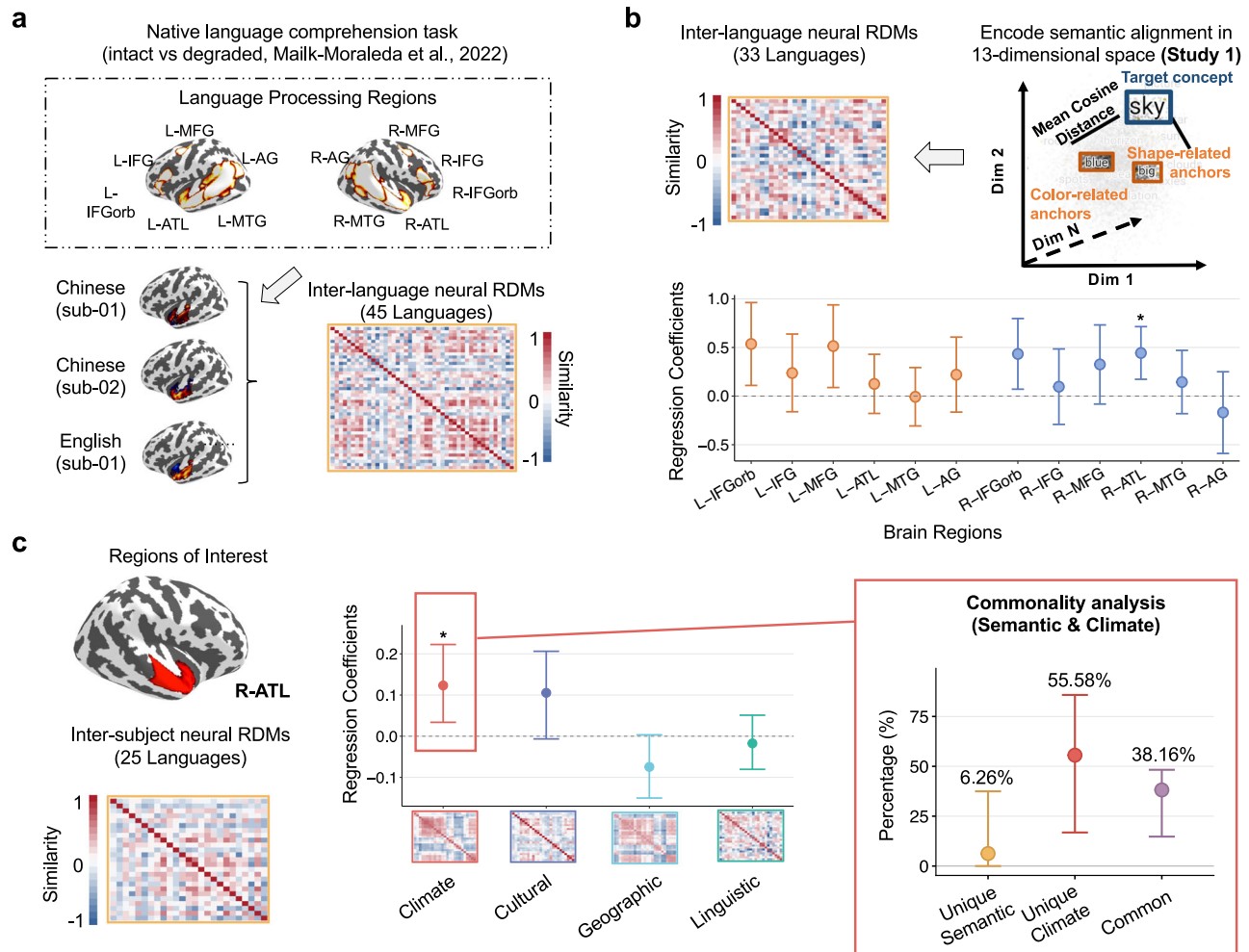

**Fig. 4 | Neural correlates of environmental effects on language processing (Study 3). a** Methodology for constructing neural RDMs from 86 participants across 45 languages within the high-level language processing network[52]. Inter-language neural RDMs were derived from the contrast between intact and degraded native language processing[51], with data averaged for each language. **b** Linear mixed regression analysis using semantic relation patterns (from the 13-neurocognitive dimensional space in Study 1) to predict cross-language neural variations. Points represent standardized regression coefficients across all 12 brain regions, derived from N = 33 languages (528 independent language pairs). The r-ATL is the only region showing significant beta values after multiple corrections (FDR-corrected P = 0.016), indicating that this region encodes cross-language semantic alignments in the 13-dimensional space derived in Study 1. **c** Climate effects on neural activity patterns in the r-ATL. Environmental factor RDMs were obtained for participants' native languages. Left panel: Points represent standardized regression coefficients for environmental predictors, derived from 25 languages (N = 300 language pairs). Right panel: The commonality analysis (right panel) quantifies the proportion of variance explained by different components. The common variance (38.16%, 95% CI [14.75%, 48.30%]) indicates that climate effects on neural activities in the r-ATL partially reflect semantically related neural activation patterns. ATL anterior temporal lobe, IFG inferior frontal gyrus, IFGorb inferior orbital frontal gyrus, MFG middle frontal gyrus, MTG middle temporal gyrus, AG angular gyrus. Error bars represent 95% CIs. Two-sided tests from linear mixed-effects models: *, P < 0.05.

stronger semantic relations. The associated semantic space for Climate PC1 and Climate PC2 is visualized for each concept (Fig. S8) and summarized by domains (Fig. 5b). For Climate PC1 (cold/temperate vs. tropical) dichotomy, concepts in general tend to exhibit higher intensity on emotional and sensorimotor dimensions (touch, motor, shape, color) in the cold/temperate zones and higher intensity on social-cognitive (social, space, number, cognition) and smell dimensions in tropical zones. For Climate PC2 (oceanic vs. continental) dichotomy, concepts tend to exhibit higher intensity on the smell, cognition, and time dimensions in oceanic zones and higher intensity on the social and sound dimensions in continental zones.

One data pattern emerged from this visualization (Fig. 5b): The dimensional difference patterns associated with climate groups are overall coherent across various concepts (and concept domains). For Climate PC1, cold/temperate climate is associated with higher loading on emotion (and sensorimotor) not only for specific concepts relating to temperature, such as *sun* or *warm*, and tropical climate is associated

with higher loading on olfactory not only for *flower* and on social not only for *father*, but for all domains of concepts in general. This pattern of association with dimensions instead of concept domains is also present for Climate PC2, although less clear-cut (see examples in Fig. 5b, lower panel).

## Discussion

Research on how the human brain processes semantics, using a handful of commonly studied languages (e.g., English, Italian, Chinese), has revealed semantic representation along a set of neurocognitive-related salient dimensions[24–27]. We tested the commonalities and variations along this neurocognitive structure across different languages using multiple approaches, including language computation models, human subjective behavioral ratings, and multi-language fMRI data (see summary in Fig. 1). Two key results were obtained. First, across 53 languages, representation on the neurocognitive semantic model had greater similarity than on alternative

**Table 2 | Environmental predictors of semantic distance across four datasets**

| Dataset | Climate | | Cultural | | Geographic | | Linguistic History | |
|---|---|---|---|---|---|---|---|---|
| | β [95% CI] | P Value | β [95% CI] | P Value | β [95% CI] | P Value | β [95% CI] | P Value |
| Word embedding data (Wiki + C) | 0.28 [0.19, 0.37] | <0.001 | 0.07 [−0.01, 0.14] | 0.097 | 0.05 [−0.03, 0.13] | 0.237 | 0.22 [0.16, 0.28] | <0.001 |
| Word embedding data (Wiki + Subs) | 0.44 [0.33, 0.58] | <0.001 | 0.22 [0.12, 0.33] | <0.001 | −0.18 [−0.29, −0.06] | 0.002 | 0.28 [0.20, 0.35] | <0.001 |
| Behavioral data | 0.53 [0.26, 0.80] | <0.001 | −0.07 [−0.36, 0.25] | 0.640 | −0.22 [−0.40, −0.04] | 0.025 | 0.13 [−0.17, 0.39] | 0.336 |
| Neural data (r-ATL) | 0.12 [0.03, 0.22] | 0.010 | 0.11 [0.008, 0.21] | 0.047 | −0.07 [−0.15, 0.002] | 0.063 | −0.02 [−0.08, 0.04] | 0.611 |

β standardized regression coefficient, CI confidence interval, Wiki Wikipedia, CC Common Crawl, Subs Open Subtitles, r-ATL right anterior temporal lobe. All estimates are from linear mixed-effects models with random intercepts for language families. All variables were standardized. P values are based on two-sided t-tests from linear mixed-effects models.

semantic models (distributional local/global models and semantic feature models) and statistical control models. The shared neurocognitive representation of the 53 languages significantly predicted the colexification network relations across 2681 languages. Second, variation patterns were accommodated by the intrinsic assumptions of the neurocognitive semantic model: the variations along such structures in terms of language embedding computation, human subjective semantic rating, and neural activity patterns during language comprehension, were all significantly predicted by major macroenvironmental factors, with variables strongly affecting the sensory inputs—climate—having the most robust effects beyond those of linguistic and cultural factors. Below, we discuss these key findings in turn.

## Stronger universality of neurocognitive model compared to control models

Previous studies on cross-language semantic alignments have focused on the degree of alignments on concepts—with how much fidelity a word translates across languages (colexification; distributional relations). Take the word *rose* as an example: to what degree can words referring to the concept *rose* in different languages be translated accurately back and forth or be related to similar words? Both universality[11] and variations[12,13] have been highlighted. Aligning with the significant universality observations, we observed that there are substantial similarities across languages above chance in semantic representations constructed from all models, including the neurocognitive dimensional (both computed and subjectively rated), distributional (local and global), and psycholinguistic semantic feature models.

Critically, the current study moves beyond magnitude analyses and tests predictions about the nature of the potential universal semantic structure—What aspects of the meaning of rose are more similar? We reasoned that a promising candidate for a universal semantic structure that is coherent with how semantics are processed in the human brain—the neurocognitive semantic structure—consists of: sensory-motor dimensions (color, shape, sound, touch, taste, smell, bodily motor) and core cognitive dimensions/domains (time, space, number, mental-cognition, emotion, social). For instance, words referring to objects containing rich color information, such as *rose*, activate color perception areas (e.g., lingual/fusiform gyrus) more strongly than those that do not (e.g., kick), and the neural activity pattern in the corresponding brain region reflects color perceptual space such that the neural activity to the word *banana* would be closer to *corn* than to *strawberry* (US data[15], Italian data[24], Chinese data[25]). Note that compared to Lewis et al.[13], which also studied distributional models and showed that global distribution space (excluding within-domain neighbors as anchors) showed less universality compared to local space (within-domain neighbors as anchors), our analyses of global distribution space included all word relationships in the vocabulary space (i.e., grand global), and showed greater universality. These results, when considered together, indicate that preserving domain structure is important in capturing cross-linguistic semantic universals. These neurocognitive dimensions, gleaned from neuroimaging studies based on a few languages (see Table S1), capture similarities across much larger language samples (53 languages, spanning 10 language families including Indo-European, Afro-Asiatic, and Dravidian), most of which have not been studied in cognitive neuroscience (e.g., Breton, Bashkir, and Tatar). Importantly, the cross-language similarity for word semantic representations computed this way is significantly greater than those computed based on relational structures with global or local neighboring words, or with psycholinguistic features that were not as neurobiologically salient. The advantage of the neurocognitive model over these control models on the same dataset is not readily explained by potential limitations of the language samples (e.g., all Eurasia). Further confidence is gained from

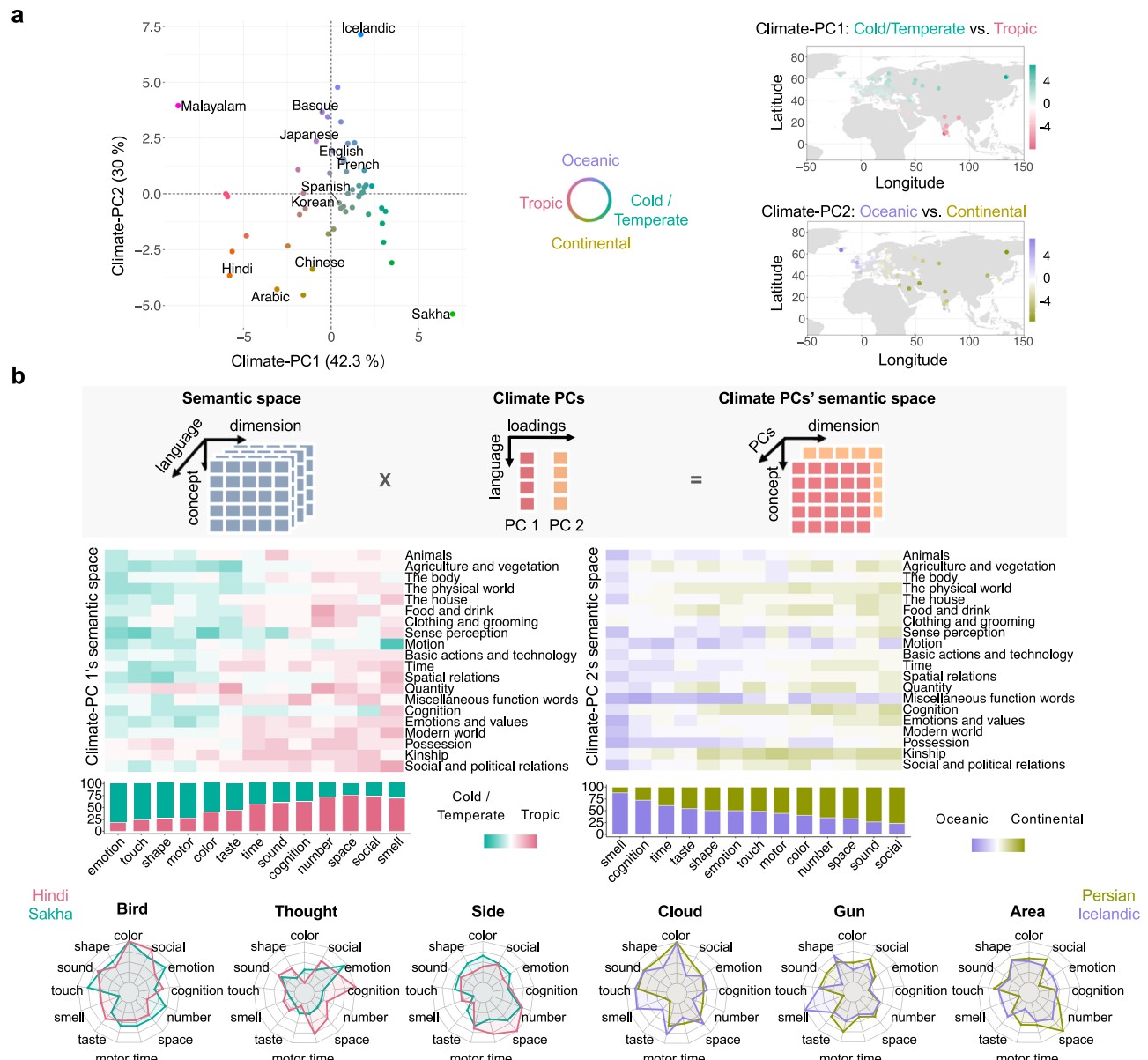

**Fig. 5 | Semantic variation structures associated with different climate groups.**
**a** Climate typology and distribution of 53 languages. PCA on the 19 climate variables of the geographic coordinates of 53 languages revealed that the first two principal components captured 72.30% of climate variations, and they were termed cold/temperate vs. tropical climates and oceanic vs. continental climates, respectively, according to the loadings of climate variables on each PC. The right panel shows the geographic mapping of climate PCs. **b** Semantic profiles associated with each climate PC. The top panel illustrates how we projected the semantic spaces obtained in Study 1 onto the Climate PC1 and PC2 axes. The middle panel shows the domain-level semantic profiles associated with each Climate PC (see Fig. S8 for the concept-level profile). Higher values along a particular direction of PCs indicate that the given climate type tends to have stronger semantic relations. The bar plots below show the semantic association ratio of the two directions along each PC, calculated as the summed values in one direction divided by the total summed values in both directions. The bottom panel shows examples of loading patterns of specific concepts on 13 neurocognitive semantic dimensions obtained from the language data.

the results showing that the shared neurocognitive representation of the 53 languages significantly predicts colexification network properties across 2681 languages. These results are readily explained by the assumption that these neurobiological structures scaffold a universal semantic representation regardless of the types of languages spoken.

## Variations along the neurocognitive semantic structure associate with climate

Regarding variations, aligning with the observations of the cultural/linguistic effects on concept-level alignment[12,13], we found that deviations along the semantic structures were significantly associated with linguistic similarity in Study 1 and cultural similarity when using another embedding model (see Supplementary Note 2), suggesting

that sociocultural processes (via word borrowing, cultural communications) could also help promote alignment on such semantic structure. The key novel finding is the robust unique effect of climate beyond these variables in explaining semantic variations across the word embedding model, human semantic behavior, and brain data.

Given the universal neural mechanism of deriving semantic representation from multimodal sensorimotor experiences (respecting core knowledge domains), environmental variables could have an effect on sensory experiences and/or related bodily functions, which in turn would affect semantic representations. Climate is one such variable. Indeed, climate properties such as temperature and precipitation contribute to human perceptual environments such as sun color, landscape variations, and interaction patterns among local

populations, plants, and animals. Recent evidence has even suggested that oxygen concentration and/or other environmental factors associated with high altitude affect color perception, which is assumed to be driven by different oxygen consumption sensitivities of different cones in the retina[53,54]. Speakers who reside in distant locations but experience similar climates are more likely to share similar sensory/perceptual inputs and thus more similar semantic representations. Notably, our results from both word embedding and behavioral rating experiments demonstrate that climate exerts a robust effect across all semantic dimensions, encompassing not only sensorimotor properties but also higher-order domains such as mental-cognition, time, space, number, social, and emotion (Fig. 3c and f). Sensorimotor dimensions are expected to be directly shaped by environmental inputs. While higher-order cognitive dimensions are further abstracted from immediate perceptual input, they arguably remain partially grounded in sensorimotor experience[55]. These findings are consistent with prior evidence that environmental conditions can modulate temporal perception[56], and that climate-linked subsistence patterns shape social behaviors and emotional tendencies[57,58], and they clearly invite further investigations into the underlying neurocognitive mechanisms. This broad dimensional effect is also in line with our intriguing observation of climate-brain activity associations in the r-ATL in the human brain. Note that we analyzed brain activity for the "intact vs. degraded" contrast, which captures broad linguistic processes including, but not limited to, semantics. Regarding the ATLs' role in language processing, accumulating research has established them as key regions for processing higher-order semantics, binding the distributed semantic dimensional representations and word relational structures in language (see reviews[26,30,59]). Indeed, the commonality analyses revealed significant shared components between climate and semantics, linking ecological variables and language neural processing that converges with our behavioral and computational findings. We do not exclude other potential language processes that may also mediate the climate-ATL association—our commonality analysis indeed also revealed significant variance shared by climate and neural activities that is not shared with semantics.

Our findings have broad implications for understanding cultural evolution. Previous research has emphasized the broad impacts of ecological variables (e.g., climate) on human sociocultural behaviors[40,41]. Given that information is ultimately processed by the internal models of the human brain for behavior, the variations along the neurobiological semantic structures may serve as a cognitive mechanism underlying such processes. Indeed, we identified specific semantic profiles that were associated with different major climate groups (Fig. 5b). For instance, the social semantic dimension loads more strongly in languages of tropical regions, which may help explain the previous reports about higher temperatures associating with stronger collectivism[40]. However, evidence regarding climate's influence on within-dimension semantic variations shows mixed patterns— Brown's findings[60] on arm-hand lexical distinctions in colder regions support climate-based accounts, while spatial reference frame variations (absolute vs. relative) may be more influenced by a complex set of factors rather than by a single factor such as climate[3]. More generally, having identified the robust association between climate and a central cognitive component—semantics, future studies elucidating the detailed neurocognitive mechanisms would be critical for understanding the intricate interplay between our living environment and diverse human behaviors, especially in the era of profound climate change.

## Limitations

A few caveats and future directions warrant discussion. First, our selection of dimensions in the neurocognitive dimensional structure, while based on existing positive findings regarding cognitive neural semantic organizations, may not be exhaustive or entirely independent. Second, true translation equivalence across languages is impalpable given theoretical and empirical considerations. We tested words from the NorthEuralex database, which contains dictionary-compiled translations. As Dellert et al.[61] noted, many languages make lexical distinctions absent in English (e.g., differentiating *air* as a breathable substance versus a space for flight), and these mappings lack extensive native speaker validation. Despite these inherent limitations, our study focuses on the underlying principles governing such cross-language (mis)alignments. Third, each method has different measures of semantics and different advantages or caveats. The statistical comparisons of the multiple semantic models in Study 1 relied on language text models, which may be sensitive to corpus sizes and text types, and semantic representations derived from text computations tend to underestimate variations stemming from non-linguistic factors[62]. The effect size varied across studies, with the variance being explained by environmental variables being higher at the language-level and much smaller in the human brain activity studies, which might be attributable to many other variables contributing to different measurements. The neural findings warrant replication with larger samples, given the limited sample size per language in the current dataset. The convergence across these different measures highlights the robustness of the positive findings we discussed. Future studies should employ tasks specifically designed to isolate semantic processing across diverse languages to more precisely characterize climate-semantic neural relationships. Finally, while our language samples included in the environment-semantic association analyses cover a substantial portion (approximately 1/2–1/3) of the world's major populations across the three studies, they have all undergone varying degrees of modernization. This may lead to an overestimation of the climate effects and an underestimation of the effects of cultural and linguistic factors on semantic structures, which warrants further investigation. Future research with more globally representative language samples is necessary to validate the generalizability of our findings.

In conclusion, we showed that the semantic representations along the neurobiologically motivated dimensions show greater alignments across diverse languages than control models, with variations along such a structure most strongly predicted by the climate in the region of the corresponding language. These findings highlight the interplay between the biological and cultural evolution mechanisms that underlie semantic meanings across the globe.

## Methods

This research complies with all relevant ethical regulations. Study 2 involved an online study conducted via a crowdsourcing platform and was approved by the Institutional Review Board of the State Key Laboratory of Cognitive Neuroscience and Learning, Beijing Normal University (approval number: IRB_A_0202_2021001). Studies 1 and 3 involved computational analyses of publicly available data and did not require additional ethical approval. No statistical methods were used to predetermine sample sizes for Studies 1 and 2. Study 3 analyzed an existing dataset with sample size predetermined by the original study. Investigators were not blinded to group assignments, as language identity was a key variable of interest across all studies. Unless otherwise specified, all statistical tests were two-sided.

### Study 1: language computational study using word embedding data in 53 languages

**Language samples and concept list.** We investigated 1016 concepts across 53 languages from 10 distinct language families. The concept list and languages were sourced from the NorthEuralex (NEL) dataset[61]. The NEL dataset provided translational word forms for these concepts, which have enduring and relatively consistent word representations in history and cover important semantic fields[12]. Language sample selection proceeded in two steps: First, we identified 61 languages with

translated word forms available in the multilingual pre-trained word embedding models (see below). Second, we excluded 8 languages missing over 25% vector representations for NEL concepts to ensure adequate shared concept coverage between language pairs. Analyses on all 61 languages yielded the same results. Detailed language information can be found in Table S2, and the distribution of concept numbers for each language is illustrated in Fig. S9.

**Multi-language pre-trained word embedding model.** We obtained semantic representations of concepts using 300-dimensional word vectors from pre-trained embedding models available through fastText (https://fasttext.cc/docs/en/crawl-vectors.html). These models were trained on vast corpora from Wikipedia and Common Crawl, with the training data containing hundreds of millions to billions of word tokens for each language, resulting in improved performance for low-resource languages compared to Wikipedia-only models (see model details[42]). FastText represents words as a combination of whole-word and subword (character n-gram) embeddings, which allows for better handling of morphologically rich languages and out-of-vocabulary words[63]. To validate our results against different corpora types, we also ran replication analyses using additional pre-trained models based on Wikipedia and Open Subtitles corpora[64], which incorporate speech transcriptions from television shows and movies (Supplementary Note 2).

**Construction of different semantic representation models.** To compare semantic representations across languages, we employed an anchor word approach[43,44]. This approach involves projecting the original 300-dimensional embedding vector representations of 1016 NEL concepts onto various anchor vectors to construct different semantic models (i.e., computing cosine similarities between NEL concept vectors and different anchor vectors). We excluded words appearing in both target concepts and anchor dimensions to avoid overestimating commonalities. We constructed the following types of models:

**Distributional semantic models (local and global).** Following Thompson et al.[12], we projected concepts onto their relationships with other words. We considered two types of measures: distributional local models, using varying numbers of semantic neighbors as anchors and distributional global models, using all the other 1015 concepts as anchors. We computed cosine similarities between each NEL concept vector and these anchors to obtain distributional semantic representations. This approach preserves the complete representational geometry within each language's native embedding space. For local models, results using the top 100 closest neighbors are reported in the main text.

**Semantic feature model.** We projected concepts onto human-generated descriptive feature words from the Semantic Feature Production Norm[33]. This norm includes approximately 4000 feature words used to describe 4436 concepts, representing various types of semantic knowledge (e.g., "is red", "is mammal", "lay eggs", and "live in the water"). As the norms were primarily generated by native English speakers, we translated key feature words into other languages in our samples using Google Cloud Translation API (https://cloud.google.com/translate), excluding Bashkir, Breton, and Sakha due to resource limitations. The semantic feature representation was obtained by computing cosine similarity between each NEL concept vector and the feature word vectors. We conducted analyses using varying numbers of the most frequently nominated feature words (top 100, 200, 500, and 1000) as anchors, with results from the top 100 feature words reported in the main text.

**Neurocognitive semantic model.** The neurocognitive semantic model consisted of 13 primary structural dimensions gleaned from

neurocognitive semantic research (see "Introduction" and Table S1 for a list of example empirical references). These dimensions include seven sensory-motor domains (color, shape, taste, smell, sound, touch, and bodily motor) and six core cognitive domains (time, space, number, mental-cognition, emotion, and social). The anchor words for each dimension were selected from the NEL concept list by three native Chinese speakers, who first manually selected words defining or corresponding to each dimension and then reached consensus on a final anchor word list. A total of 122 anchor words were used (see Table S3), with approximately 2–14 anchor words for each dimension. We computed the averaged cosine similarity between each NEL concept vector and the vectors of anchor words for each dimension to obtain a 13-dimensional vector as the neurocognitive semantic representation. Using the original anchor concepts from Binder et al.[32], which were selected by English speakers, yielded the same result patterns.

**Statistical control models.** To assess the explanatory power of the above candidate theoretical models for cross-linguistic semantics commonalities, we constructed two statistical control models: (a) Random word model. We randomly selected 13 words from the original 1016 NEL concepts as anchor words and calculated their cosine similarity with the target concepts. (b) Random dimension model. We randomly selected 100 anchor words, grouped them into 13 dimensions (corresponding to the number of dimensions of the neurocognitive model) using K-means clustering, and calculated their cosine similarity with the target concepts. This random dimension model provides a dimensionality-matched control, demonstrating that larger cross-language alignment is not simply caused by having fewer dimensions. Each random model was repeated 10,000 times to obtain the cross-language commonality distributions. We then established the statistical confidence of each theoretical model by comparing the model's cross-language commonality scores with the distribution from the statistical control models.

**Estimation of cross-language commonalities.** To assess the degree of cross-language commonality across different semantic models, we employed two complementary measures of convergence. Note that these measures also differ in the number of concepts analyzed and their treatment of between-concept variations.

**Inter-language correlation analysis.** For each semantic representation model, we calculated the Pearson correlation coefficient (r) between the semantic representations of each language pair for all 1016 concepts. The resulting inter-language correlation (ILC) was then Fisher's r-to-Z transformed and averaged across concepts. To compare the ILCs between different types of theoretical semantic representations, we employed the Wilcoxon signed-rank test (two-sided) for paired samples.

**Principal component analysis.** To further estimate shared variances across all languages, we conducted principal component analysis (PCA) for 204 concepts shared across all 53 languages. For each semantic representation model, we represented each language as a matrix of semantic relations (e.g., for the neurocognitive model, the matrix dimensions were $204 \times 13$). We then treated languages as features and semantic relations (reshaped from a matrix into a vector) as samples, and performed PCA across languages. The proportion of variance explained by the first principal component (PC1) was used as an estimate of shared variance for cross-language semantic alignments.

**Prediction of colexification network topologies.** To investigate whether the neurocognitive semantic models could predict word relations in more diverse language samples, we employed the

Database of Cross-Linguistic Colexifications (CLICS version 3.0), a comprehensive cross-linguistic resource containing colexification patterns of approximately 3000 concepts across 2681 languages in 180 language families[48], capturing instances where two or more concepts are expressed by the same wordform within a language. The hypothesis here is that if the neurocognitive semantic model outperforms statistical control models (random word models and random dimension models) in predicting word topological patterns in the colexification network, it would indicate the generalizability of these structures to a broader range of languages.

We conducted separate analyses using colexification networks derived from: (a) The entire CLICS language sample ($N = 2681$); (b) A subset of non-Eurasian languages from the CLICS database. For 165 of the 204 concepts used in the pre-trained embedding analyses, we extracted the following network properties from the CLICS database: edge weight, weighted common neighbors, and community structures based on Louvain and Infomap algorithms. These measures capture first-order, second-order, and global information in the network, respectively, and are widely used to quantify semantic similarity across languages[14,65,66]. We computed an overall pairwise similarity score as the average of these four metrics and then correlated these network similarities with the neurocognitive semantic relations from the 53 languages in our embedding data using Spearman correlation. The correlations were averaged across the 53 languages to obtain a mean correlation for each network (full sample and non-Eurasian subset). The statistical confidence of the neurocognitive models' predictability for colexification network properties was established by comparing the model's Spearman correlation coefficients with the distribution of correlation coefficients from the statistical control models ($N = 1000$ times).

**Prediction of inter-language semantic variations.** To understand what environmental factors may account for the variations in the universal semantic structure, we carried out representational similarity analysis (RSA) between inter-language semantic variation and the distances of the four salient ecological variables, including the geography, climate, linguistic history, and culture.

**Environmental variables.** We obtained pairwise distances on these four environmental variables based on the languages' coordinates provided by Glottolog 4.6[50]. The geographic distances were calculated as the geodesic distances between the locations of languages on the Earth's surface. The climate distances were calculated as the scaled Euclidean distance based on the estimates of 19 bioclimate variables from WorldClim[67] (see Table S7 for the full list). The 19 bioclimate variables were derived from the monthly temperature and precipitation, which are often used in species distribution and related ecological modeling. The distances of linguistic history were estimated based on cognates of the NorthEuralex wordlist, using LingPy[68] with the LexStat method[69]. The cultural distances were retrieved from Thompson et al.[12], which were calculated based on the eco-cultural traits in the D-place dataset[70], including societies' housing, labor institutions, marriages, and political systems.

**Representational similarity analysis.** The inter-language RDMs were first constructed for semantic variations and for each environmental variable. The semantic RDM was constructed by computing the distance (i.e., 1−Pearson correlation, with Fisher's r-to-Z transformed) of averaged concepts' neurocognitive semantic vectors between language samples. After obtaining these inter-language RDMs, we performed nonparametric Spearman correlations for their raw associations and built multiple regression models to estimate how the semantic variations across languages could be explained by the environmental variables. Specifically, we tested the association between the lower triangle of the semantic representational

dissimilarity matrix (RDM) across languages and the lower triangle of RDMs for each or composite of these environmental variables (excluding the main diagonal elements). The linear mixed regression model was further conducted to estimate the fixed effects of each environmental variable with a crossed random-effects structure that nested language pairs within language families considered, to account for potential non-independent language sampling[12,71]. The model can be formally expressed as:

$$\text{Semantic\_Distance}_{ij} = \beta_0 + \beta_1 \text{Climate}_{ij} + \beta_2 \text{Culture}_{ij} + \beta_3 \text{Geography}_{ij}$$
$$+ \beta_4 \text{Linguistic\_History}_{ij} + (1|\text{Family}_i) + (1|\text{Family}_j) + \varepsilon_{ij}$$

$$(1)$$

where $i$ and $j$ index the language pairs, $\beta_0$ represents the intercept, $\beta_1$ through $\beta_4$ represent the fixed effects for each environmental variable, $(1|\text{Family}_i)$ and $(1|\text{Family}_j)$ represent the random intercepts for the language families of languages $i$ and $j$ respectively, and $\varepsilon_{ij}$ represents the residual error term (see Supplementary Note 3 for details and other ways to control non-independence). Effects of environmental factors on cross-linguistic semantic variations at the single dimension level were also assessed through linear mixed regression models.

**Study 2: human participants' behavioral ratings in 8 languages**
**Participants.** We recruited a diverse sample of 272 online participants from 58 city sites across 8 countries, representing distinct languages: Arabic (Egypt), Chinese (China), English (USA), Hindi (India), Japanese (Japan), Korean (South Korea), Russian (Russia), and Spanish (Spain). Participants were recruited through the Appen crowdsourcing platform (http://www.appen.com) and provided informed consent prior to participation. Participants were compensated for their time through the Appen platform according to standard platform rates. All participants were confirmed to be native speakers residing in their respective targeted sites. To ensure geographical diversity while minimizing intercorrelations among environmental variables of interest, we strategically selected 6–8 geographically dispersed sites within each country (Fig. S6 and Table S5). To ensure data quality, we implemented a stringent inclusion criterion. Participants whose rating vector correlated less than 0.5 with the averaged group vector within their respective language were excluded from the main analyses. This resulted in the exclusion of 19 participants. The final sample size was 253 participants (self-reported gender: 113 male, 139 female, 1 other/ preferred not to say; age range: 18–83 years), with a balanced distribution of 30–34 participants per country. Gender and age were included as covariates.

**Concept list.** For this study, we used the Swadesh 207-Word list, which has been extensively studied in historical and comparative linguistics and is considered to represent the core basic vocabulary of human languages[72]. The 207 words also sufficiently overlapped with the NEL concept list used in Study 1. To obtain translated word forms of these concepts across our target languages, we sourced initial word forms from the PanLex Swadesh database[73] and asked professional translators to review them. In cases where PanLex provided multiple word forms for a single concept, our translators were instructed to select the most commonly used ones. To mitigate potential ambiguity, particularly for polysemous words, we accompanied them with contextual specifications (e.g., lie (as in a bed)). The final word forms are provided in Table S6.

**Rating instructions.** To evaluate the 13-dimensional neurocognitive semantic structures, participants were instructed to rate concepts on the extent of association with specific dimensions. Our approach was adapted from Binder et al.[32], providing some forms of association for each dimension (e.g., for color, the association could be "This word refers to something that has a characteristic color (e.g., eggplant)",

"This word describes the change in color (e.g., fade)", or "this word directly refers to a particular color (e.g., red)"). To support participants' understanding, we included example words with high and low loadings from Binder et al.'s questionnaire. To mitigate potential cross-cultural biases, we excluded example words with below-average frequency based on word frequency data in Binder et al.[32] and example words that could introduce cultural biases. The English rating instructions were translated into 7 other languages using multi-proofreading processes (Supplementary Note 6). The rating instructions for each language can be found at the link (https://osf.io/suyeb).

**Online rating procedures.** Participants completed a prescreening process followed by three sessions of the conceptual rating task.

**Prescreening.** To ensure native speaker selection, we employed a rigorous prescreening process comprising self-reports and a 5-min audio transcription test in the target language. We collected detailed language background and demographic information, including age, gender, education level, ethnicity, subjective socioeconomic status[74] (SES), second language proficiency, country of upbringing before age of 7, and current region of residence. Participants who were non-native speakers, failed the transcription test, had lived in a foreign country before the age of 7, or were currently residing in foreign countries were not invited to the rating task. Moreover, those who self-reported high proficiency in a second language (5 on a 5-point Likert scale, indicating "upper intermediate and above") were also excluded. For Indian participants, criteria were adjusted due to the country's multilingual nature, excluding only those with high proficiency in languages of interest (e.g., English).

**Rating task.** Eligible participants completed a rating task comprising three 50-min sessions. Each session involved rating approximately 70 words on 13 neurocognitive dimensions. The 207 concepts were pseudo-randomly shuffled into 30 wordlists, which were then randomly assigned across the three sessions. To ensure quality control during the process, three catch trials using high-loading examples from the instructions were randomly inserted into each wordlist. The task automatically terminated if erroneous (low) ratings were detected on control questions. Pearson correlations were computed between each participant's rating vector and the averaged group vector within each language. Participants with correlations below 0.5 were excluded from further analysis. The final sample consisted of 253 participants. Analyses including all participants yielded similar results.

**Prediction of cross-language speakers' semantic variations.** An inter-subject correlation matrix was constructed by calculating Pearson correlation coefficients between each participant pair's ratings on 207 concepts across 13 dimensions (2691 ratings per participant). We first analyzed the variance components shared across individuals (universal), across languages (language-specific), and individual errors to validate the commonality results (see Supplementary Note 4). To investigate the explanatory power of macro environmental variables in understanding cross-language (neurocognitive) semantic variations, we then shifted our analysis to the language-level and employed RSA to predict language-level semantic variations using the distance of the four environmental variables, including climate, geography, linguistic history, and culture.

**Environmental variables.** We obtained environmental variables for each language by aggregating information from participants' geographic locations. Specifically, we first collected city/county location data from all participants within each language group. The site-level geographic coordinates were obtained using the Bing Maps API (https://www.bingmapsportal.com/). For each language, we then calculated the average coordinates to represent its geographic center.

The climate profile for each language was derived by extracting the 19 bioclimate variables from WorldClim[67] at these averaged coordinates. The geographic distances between languages were calculated as the geodesic distances between these averaged coordinates. The linguistic history and cultural distances were taken from the language/country level distance measurements from Study 1. Additionally, we constructed a measure of demographic distances based on scaled Euclidean distances to account for potential confounds (age, gender, education level, and SES) between language groups.

**Representational similarity analysis.** For language-level analysis, we first computed average ratings for each concept-dimension pair within each language to create language-level semantic representations. The semantic RDM was constructed by computing the distance (1−Pearson correlation, with Fisher's r-to-Z transformed) between these language-level semantic representations. We then constructed language-level environmental RDMs for climate, culture, geography, and linguistic history, as well as a demographic RDM.

We first assessed raw association patterns between the semantic RDM and environmental RDMs using nonparametric Spearman correlations. We then conducted hierarchical regression with demographic distance entered as the first step, followed by the four environmental variables in the second step to evaluate their collective contribution beyond demographic differences in explaining semantic variations. Following the same procedures as in Study 1, we conducted linear mixed regression models to estimate the unique fixed effects of each environmental variable while controlling for the others. This model included all environmental variables as simultaneous predictors with a random effects structure that included random intercepts for language families to account for phylogenetic relationships:

$$\text{Semantic\_Distance}_{ij} = \beta_0 + \beta_1\text{Climate}_{ij} + \beta_2\text{Culture}_{ij} + \beta_3\text{Geography}_{ij} + \beta_4\text{Linguistic\_History}_{ij} + (1|\text{Family}_i) + (1|\text{Family}_j) + \varepsilon_{ij}$$

$$(2)$$

Effects of environmental factors on cross-language semantic variations at the single dimension level were also assessed.

## Study 3: human participants' brain activity patterns in 45 languages

**fMRI dataset.** This study utilized an open-access multi-language fMRI dataset[51] available at the OSF repository (https://osf.io/cw89s). The dataset comprises neural activity measurements during a language comprehension task performed by 86 native speakers across 45 languages from 12 language families. Eighty-six participants (43 reported as males; age range: 19–45 years) were recruited from Boston, the United States. Sex/gender was balanced, and no explicit sex- or gender-based contrasts were performed in our secondary analyses of this dataset. All participants were proficient in English in addition to their native language. Participants underwent a passive listening task during fMRI scanning. They were presented with auditory stimuli consisting of passages from "Alice in Wonderland" translated into their native languages. The stimuli were presented in two conditions: intact native-language condition and acoustically degraded-language condition (control condition). This contrast between the two conditions is commonly used to obtain neural activities that are related to high-level language processing, including semantics. We obtained the pre-processed whole-brain contrast maps (t-value maps comparing intact native-language and acoustically degraded-language conditions) for all 86 participants from the OSF repository for our analysis. Detailed information on participant characteristics, imaging acquisition procedures, task designs, data preprocessing and first-level modeling are referred to Malik-Moraleda et al.[51].

**Prediction of cross-language speakers' neural variations**. We investigated whether environmental effects were reflected in cross-language speakers' neural activity patterns during language processing using RSA methods. Our analysis proceeded in the following steps:

**Identification of brain regions of interest (ROIs) encoding cross-language semantic alignment**. For each of the 12 language-responsive regions, we first aggregated individual neural responses at the language-level by averaging $t$-value maps from speakers of the same language. We then assessed whether neural activity patterns in these regions reflected cross-language alignment in the 13-neurocognitive-dimensional semantic space derived from Study 1. For this analysis, we constructed neural RDMs for each brain region and compared them with semantic RDMs using a linear mixed regression model that included random intercepts for language families:

$$\text{Neural\_Distance}_{ij} = \beta_0 + \beta_1 \text{Semantic\_Distance}_{ij} + (1|\text{Family}_i) + (1|\text{Family}_j) + \varepsilon_{ij}$$
$$(3)$$

**Construction of environmental variables**. We obtained environmental variables for each language based on Glottolog 4.6 coordinates[50]. For these coordinates, the climate distances were calculated as the scaled Euclidean distance based on the 19 bioclimate variables from WorldClim[67]. The cultural distance measures were taken from Study 1. The geographic distances were calculated as the geodesic distances. The language history distances were obtained from lang2vec (https://github.com/antonisa/lang2vec), which calculates the distance between two languages by counting the number of upward steps required on the tree until both languages converge at a common node, circumventing the need for cognate calculations on specific wordlists.

**Representational similarity analysis**. To assess environmental influences on neural activity patterns in the r-ATL, we constructed language-level environmental RDMs for climate, culture, geography, and linguistic history. We then used a linear mixed regression model to test whether these environmental variables predicted neural pattern dissimilarities:

$$\text{Neural\_Distance}_{ij} = \beta_0 + \beta_1 \text{Climate}_{ij} + \beta_2 \text{Culture}_{ij} + \beta_3 \text{Geography}_{ij}$$
$$+ \beta_4 \text{Linguistic\_History}_{ij} + (1|\text{Family}_i) + (1|\text{Family}_j) + \varepsilon_{ij}$$
$$(4)$$

**Relationship between semantic and environmental effects**. To further characterize the relationship between neural activity, semantic and environmental variations, we conducted a commonality analysis. This analysis partitioned the variance in neural pattern dissimilarities into components uniquely attributable to semantic factors, uniquely attributable to environmental factors, and shared between them. This analysis provided insight into how environmental and semantic factors might jointly influence neural representations during language processing.

#### Reporting summary
Further information on research design is available in the Nature Portfolio Reporting Summary linked to this article.

## Data availability
The multilingual fastText embeddings used in Study 1 are publicly available at https://fasttext.cc. Additional distributional semantic vectors used in validation analyses were obtained from the subs2vec model (https://github.com/jvparidon/subs2vec). The NorthEuraLex word list and concept translations are available at http://northeuralex.org.

Colexification data were obtained from the CLICS database (version 3.0; https://clics.clld.org). Climate variables were obtained from the WorldClim 2.0 database (https://www.worldclim.org). Cultural variables were obtained from the D-PLACE database (https://d-place.org). The behavioral rating data from Study 2 (13-dimensional ratings for 207 concepts in 8 languages), together with derived semantic matrices and environmental distance matrices, are available at OSF. The fMRI data re-analyzed in Study 3 were originally collected and shared by Malik-Moraleda et al.[51] and are available at OSF. Our derived language-level neural representational dissimilarity matrices and analysis scripts are available at OSF. Source Data underlying the main figures and tables are provided with this paper. Source data are provided with this paper.

## Code availability
All data analyses were conducted using Python (version 3.9.1) and R (version 4.5.2). Complete details of software packages and analysis code are available at OSF.

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

## Acknowledgements

We thank Haojie Wen, Xi Yu, Shuang Tian, Dingchen Zhang, and Shuyue Wang for help and comments on earlier drafts of the manuscript. We are grateful to Yongqin Liu for insightful discussions on climate-related analyses. This work was supported by the Brain Science and Brain-like Intelligence Technology - National Science and Technology Major Project 2021ZD0204100 (2021ZD0204104 awarded to Y.B.), and the National Natural Science Foundation of China (32595491 awarded to Y.B., 32171052 to X.W.).

## Author contributions

Y.B. conceived and supervised the research; Z.F. performed the research; Y.C., T.Z., and Y.L. collected rating data; Z.F. and Y.C. analyzed the data; X.W. contributed valuable discussions; and Y.B., Z.F., and X.W. wrote the paper.

## Competing interests

The authors declare no competing interests.
