## [Transparent Peer Review file · Nature Communications]

Semantic similarity across languages reflects neurocognitive dimensions shaped by climate

Corresponding Author: Professor Yanchao Bi

Version 0:

Reviewer comments:

Reviewer #1

(Remarks to the Author)

I have seen previous versions of this manuscript, and I approved them, as far my expertise is concerned, so I also approve this version. Data and code are much clearer than before, minor points from my side were addressed. Bigger concerns regarding the approach and conclusions that are beyond my expertise may be raised, but it is beyond my abilities to bring them up here. So I agree with the publication of the manuscript.

(Remarks on code availability)

After previous discussions with the authors, they have tried to comply with my requirements to provide replicable code, specifically providing an extensive README file, but also program requirements, and instructions of how to run the experiments on one's own machine. I am happy with what I see here, but I have to admit I did not actively RUN the code for time constraints, as Nature Comms only gives me two weeks for the reviews and sends annoying emails to remind me to make them earlier.

Reviewer #4

(Remarks to the Author)

Summary:

Fu et al.'s study tackles two main questions:

How well do different semantic frameworks capture cross-lingual alignment in meaning?

What variables affect inter-language differences in alignment?

To address question (1), the authors compare their primary candidate, i.e., the "neurocognitive framework" (henceforth NF, which represents meaning as loadings onto 13 cognitively relevant semantic dimensions, such as color, shape, time, space), with a) distributional semantic models, b) semantic feature models, and c) statistical control models. They find that the NF better captures cross-lingual alignment in semantics, as shown by analyses on word embeddings in multiple languages and a colexification network.

To address question (2), the authors use representational similarity analysis, assessing the second-order similarity between "semantic" RDMs (based on word embeddings, human ratings, and fMRI data) and RDMs based on geographic, linguistic, cultural, and climatic information. The core finding is that climate-based RDMs explain cross-lingual variation in semantic structure.

Evaluation:

We find the hypothesis that climate might modulate cross-lingual differences in semantics intriguing and the motivation for investigating this question is convincing: climate strongly modulates sensory signals across modalities, and given that meaning is tied (at least in part) to sensory information, it follows that climate is a reasonable candidate for modulating semantic idiosyncrasies. We also commend the methodological breadth of the article, spanning computational modelling,

behavioral experimentation, and neuroimaging; and we find the baseline conditions to be strong (with two complementary statistical control models). Nonetheless, we have serious concerns about the methodological approaches adopted in the various studies, which need to be addressed.

The main concern

Representational similarity analyses. The most substantial concern we have is with how representational similarity analyses are conducted across the studies. In the three studies assessing the impact of climate on semantics (Study 1B, 2, and 3), the authors adopt ad-hoc statistical testing where each dependent variable is modeled with a different statistical approach: In Study 1B, the authors predict NF-based RDMs for individual languages with RDMs based on climate, geographic, linguistic history, and culture variables. The analysis involves a crossed random-effects structure that nests language pairs within language families.

In Study 2, the authors construct RDMs for individual participants. Unlike in Study 1B, they convert the climate RDMs to ranked data. The mixed effects models involve subject pairs nested within language families (but not languages).

In Study 3, the authors use a hierarchical approach by which they compare nested models: first, a “baseline” model incorporating geographic distance and linguistic family as phylogenetic controls, and then a target model that additionally includes climate and cultural distances. The RDMs are constructed at the level of individual participants. The random effect structure only includes random intercepts for each participant dyad. The nested models are compared by means of ΔAIC . The three approaches are very different: sometimes the distance matrices are converted to ranked data (Study 2), the random effect structure of the mixed models varies from study to study, and while Study 1B and Study 2 simply examine the joint influence of several variables on the outcome variable, Study 3 adopts a model comparison approach. Most critically, the paper makes inferences about inter-language differences, but only Study 1B analyzes the data at the level of individual languages, whereas both Study 2 and Study 3 analyze the data at the level of individual participants.

If these studies are to be combined in a single study, the analytical approach should be consistent across them. One convincing approach would be to analyze the data at the level of individual languages (aggregating across participants) within a model-comparison framework, where a baseline model (including geographic, cultural, and linguistic history distances) is compared with a model that additionally includes climate, with random effects (ideally intercepts and slopes) for the languages in the pair. The key advantage of this analytic approach (where the analyses are conducted at the level of individual languages) is that it would most directly support the key claims of the paper about inter-language differences. Moreover, the approach where multiple participants from the same language are included in the RDMs may inflate the second-order similarity between climate RDMs and semantic RDMs simply because speakers of the same language have the same climate and possibly similar semantics. However, we acknowledge that other analytic solutions are possible; the only critical conditions are: (i) using the same analytic approach across studies and (ii) providing theoretical reasons why there might be (small) differences in the analyses across experiments. If the findings prove robust and come out reliably when the same analytic procedure is applied across studies, this would make us (and likely other readers) more confident that the findings are real and meaningful, and would likely increase the long-term impact of this work.

Additional major concerns

Brain activity patterns. In Study 3, the authors correlate RDMs based on the t-maps for the intact-degraded contrast (fMRI data from Malik-Moraleda, Ayyash, et al., 2022) with RDMs based on climate, geographic, linguistic history, and culture variables. We don't think that the authors characterize this result properly. For instance, in their response to Reviewer #1 (from the earlier round), they say: “We found that in the right ATL brain region, languages associated with similar climates showed similar semantics-related neural activities”. However, this is not what the t-maps for the intact-degraded contrast reflect; this contrast indicates, for each voxel, how much stronger it responds to intact vs. degraded language stimuli (this contrast is often used to identify brain areas that respond to language). This contrast does not isolate semantic processing, instead including many other aspects of language processing (e.g., phonology, morphology, syntax). What these results (if real) would indicate is that climate modulates the localization of the language areas across languages. This is a very different (and very strong!) claim. This claim is barely related to the conclusions drawn from Study 1B and Study 2, which are clearly about semantics. Moreover, the results seem highly surprising given the substantial interindividual variability in the topography of the language areas; finding meaningful interlinguistic variability on top of this interindividual variability is generally challenging, especially given that the dataset the authors draw on has only two individuals from each language.

Representational similarities and differences within the NF. The NF in Study 1 is based on projecting word embeddings onto a lower-dimensional space of 13 dimensions/features. The authors found that this lower-dimensional space better reflects semantic similarities across languages. This makes sense, as this space consists of a possibly universal set of conceptual dimensions and filters away variance along other potentially relevant semantic axes that might be more idiosyncratic and language-specific. Yet, they use this space to study cross-lingual differences and how those are modulated by climate. There appears to be a contradiction here—if the authors want to model cross-lingual differences in semantics, why choose a representational projection that minimizes those differences? One motivation for restricting the feature space to those semantic dimensions could be that, in principle, climate could be expected to modulate semantic differences tied to the sensory information conveyed by words, and for this reason, it would make sense to restrict the focus to those features. If this is the case, this line of reasoning should be clearly formulated in the article (see also the following point “Sensorimotor and cognitive dimensions”), because the original line of reasoning put forward by the authors (i.e., adopting the NF because it better identifies cross-lingual consistencies in semantics) is not convincing. More generally, the authors should emphasize the point we made above, namely that the NF restricts semantic variation by projecting meaning onto a small set of semantic dimensions. Relatedly, it would be interesting to see how much information is lost in this projection (partialling out information beyond the 13 NF axes); this could be achieved by doing RSA over the native word embedding space and the

NF space in the various languages.

Characterization of the alternative models. The authors present three alternatives to the NF: a) distributional semantic models, b) feature-based psycholinguistic models, and c) statistical control models. However, we believe that the authors' operationalization of feature-based psycholinguistic models (calculating the cosine similarity between a target word and the 100 most frequent English features in the Buchanan et al. (2019) norms) mischaracterizes how feature models are intended to be used (where each word is represented as a set of discrete features that characterize a concept, like "is red", "is mammal", etc.). After all, all the models they consider in their first study are distributional semantic models, where word embeddings are projected onto lower-dimensional spaces which consist of neighbouring words (a), frequent features (b), or randomly sampled words (c). Even the NF consists of a projection of word embeddings onto the 13 cognitively relevant features. Since all the considered models are ultimately projections of a distributional semantic model (DSM), we believe it is warranted to include the "native" representations of the DSM itself. This would require:

Evaluating how similar the different languages are in the native fastText word embeddings space (without projecting embeddings onto anchor words), and

Evaluating the similarity between the native fastText-based RDM and the RDMs based on climate, geographic, linguistic history, and culture.

Sensorimotor and cognitive dimensions. The NF uses 13 dimensions; 7 are sensorimotor (color, shape, ...), while the remaining 6 are called "cognitive domains" (time, space, number, ...). While it is rather clear why climate should have an effect on sensorimotor representations, the authors need to elaborate why climate would affect the "cognitive domains" dimensions as well.

Minor points:

Semantic projection. The authors say they employ the "semantic projection" method to project word embeddings onto the 13 dimensions of the NF. Nonetheless, their approach only consists in calculating the cosine between each word and a few anchor words for each dimensions, which does not correspond to the semantic projection method proposed by Grand et al. (2022)—in fact, it was found to perform worse than the actual method, which involves calculating a dimension vector as the difference between two "poles" (for the sound dimension they could be something on the lines of "noisy" vs. "silent"), and then projecting individual words onto these dimensions with the dot product. We understand that it might be difficult to define the poles for some features which are not intrinsically unidimensional (e.g., shape), but this should then be clarified.

Language and thought. From the abstract: "languages are used worldwide as the primary means of human thought". This is a rather controversial statement, we'd recommend the authors to tone it down in light of evidence that language and thought appear to be largely dissociable (see Fedorenko, Piantadosi, Gibson, 2024 for a recent review).

Language family distance. The language family distance metric (1 if two languages belong to the same family, 0 otherwise) is a coarse metric. The authors could consider using a distance metric based on a language tree. For instance, the genetic distance calculated by the Python package lang2vec calculates the distance between two languages by counting the number of upward steps required on the tree until both languages converge at a common node (Littell et al., 2017).

Typos and writing suggestions:

The core hypothesis of the paper (that climate might modulate cross-lingual differences in semantics because of general perceptual differences in different climatic environments) becomes clear only at ll. 130-137; we recommend to clearly specify this point earlier in the introduction so that readers can understand from the get-go what the article is about.

Throughout the article, the authors consistently refer to word embedding models as "language models"; consider rephrasing as word representation models or word embedding models. Also, fastText (the word embeddings model used in the study) is a peculiar word representation model in that it represents words as a sum of word embeddings plus subword (n-gram) embeddings. This should be mentioned somewhere in the paper.

The theoretical basis of the NF is not very clear: it starts by positing that language and sensory experience form the basis of conceptual representations (ll. 77-99) and then concludes that an appropriate way to represent semantics is by using 13 dimensions (sensorimotor dimensions and "cognitive domains"), without mentioning how language information fits within this model and where the dimensions from the "cognitive domains" come from.

I. 63: "beauty in English, bella in Italian" – it should either be "beautiful in English, bella in Italian" or "beauty in English, bellezza in Italian".

II. 119-120: "These alternative models [distributional semantic models, feature models] are prevalent in cognitive disciplines without explicit considerations of the neurobiological basis and have been shown to be less correlated with word semantic brain activities than the neurocognitive dimensional model [...]" – distributional semantic models and their successors (e.g., language models) are often linked to neurobiological principles, and obtain great fit to brain activity patterns (see e.g., Goldstein et al., 2022; Shrimpf et al. 2021).

I. 299: It would be great if the authors could write the lmer formulas to better understand the random effect structure of the mixed models (also, I. 347).

I. 352: The p-value is missing for the coefficients of "linguistic history".

II. 211-212: In the context of the finding that the "global" distributional space finds more cross-lingual commonalities than the "local" distributional space, consider citing Lewis et al. (2023), who found the opposite pattern.

l. 596: "billions to hundreds of millions" → "hundreds of millions to billions"
l. 713: "with Fisher-Z transformed" → "with Fisher's r-to-z transformation"

Questions:

Distributional local vs. global models. It is unclear how the distributional local models work. The authors say that in the local condition, each word is projected onto its immediate neighbors. We do not understand how this might work—if each word is projected onto its neighbors and the neighbors change from word to word, this would create a dimensionality mismatch (e.g., w_1 would be projected onto w_2, w_3, w_4 etc., while say w_{123} would be projected onto w_{124}, w_{125} etc., so the dimensions of the projection of w_1 would be different from the dimensions of the projection of w_{123}). Could you please clarify how this approach works? Relatedly, the global models use as anchors all the words but the target word itself. This would once again cause a dimensionality mismatch, as, for instance, the anchor w_1 would be absent from the embedding projection of w_1 , and the anchor w_{123} would be absent from the embedding projection of w_{123} .

References

Lewis, M., Cahill, A., Madnani, N., & Evans, J. (2023). Local similarity and global variability characterize the semantic space of human languages. *Proceedings of the National Academy of Sciences*, 120(51), e2300986120.
Littell, P., Mortensen, D. R., Lin, K., Kairis, K., Turner, C., & Levin, L. (2017). Uriel and lang2vec: Representing languages as typological, geographical, and phylogenetic vectors. *Proceedings of the 15th Conference of the European Chapter of the Association for Computational Linguistics: Volume 2, Short Papers*, 8–14.
Fedorenko, E., Piantadosi, S. T., & Gibson, E. A. (2024). Language is primarily a tool for communication rather than thought. *Nature*, 630(8017), 575-586.

(Remarks on code availability)

Reviewer #5

(Remarks to the Author)

Fu et al. address an important yet neglected topic—namely, crosslinguistic similarities and differences in meaning, and their implications for cognition—by combining computational linguistics and neuroscience. Their methods are innovative and their results are interesting and valuable. I do, however, have some concerns.

Like the study by Thompson et al. (2020), this one relies heavily on Dellert et al.'s (2020) NorthEuralex database, but this database has several limitations that Fu et al. should explicitly acknowledge. Two major limitations are as follows.

1. Fu et al. state that the "target concepts" and "anchor terms" they draw from this database are "translation equivalents" across the 53 languages they focus on. But this is not true. Even Dellert et al. are quite open about this. They state: "For instance, many languages lexically distinguish the substance one breathes from the space in which birds move, making it necessary to further specify the English equivalent "air". For some concepts (especially kinship terms), the specification also defines which term is to be mentioned first in our list of equivalents if a more fine-grained distinction is made in the target language, e.g. 'grandfather (e.g. father's father)'."

All of the information in the database was compiled from dictionaries, without extensive validation from native speakers. This is another shortcoming that Dellert et al. (2020) admit and that Fu et al. should make clear too.

Crosslinguistic lexical-semantic studies that HAVE included native speaker judgments have revealed nontrivial differences in how even closely related languages partition the same conceptual domains, and this has implications for Fu et al.'s study. As described below, two examples involve color terms and body part terms, but comparable results have been reported for many other domains.

For instance, Fu et al. use lots of basic color terms (e.g., blue, red, yellow, etc.) as "anchors" for their neurocognitive color dimension. But Majid et al. (2015) show that there are not straightforward translation equivalents for these terms across 12 Germanic languages. Here's a snippet of their discussion: "Within the colour domain, English and Danish differed most from one another (see Table 2 with correlations). English red is very localised: it was the dominant response for only one colour chip. In contrast, Danish rød was a dominant response for 11 different chips. In English, speakers preferentially called most of those chips pink, which was then also extended into an area that Danish speakers called lilla. Similarly, English orange was smaller in extension than its Danish equivalent orange. English yellow and Danish gul are approximately the same size, but because Danish orange has a larger extension gul seeps into territory that English speaker preferentially name green."

The body part domain is relevant to Fu et al.'s study because they used lots of body part terms as "anchors" for their neurocognitive motor dimension. But Majid et al. (2015) report nontrivial lexical-semantic differences between 12 Germanic languages for such terms. Here's part of their discussion: "...there is no single cognate set used in the Germanic languages for 'leg', and we see a more divergent pattern of naming. Participants saw red dots on the upper and lower leg, on the right and left side of the body, viewed from front and back. This lets us examine how they chose to refer to these distinct images. English speakers predominantly used leg for all such stimuli; similarly Dutch, Flemish and Luxembourgish used a single term been/Been. Frisian speakers also used been, but not as frequently. However, the speakers of the other languages tended to use distinct terms for the upper and lower leg. German speakers used Oberschenkel and Unterschenkel, literally 'upper-leg' and 'lower-leg', but the root *Schenkel alone was never attested. Schwyzerdütsch speakers used the cognate

terms, obèrschènku and ungerschènku, although bare schènku was also used (but only twice, and only with reference to upper leg stimuli). The Scandinavian languages make sporadic use of bein/ben, but for the most part other terms were more frequent. Like German speakers, the participants preferentially distinguished upper and lower leg. This discussion illustrates that even in a domain with much shared structure, there are still many intricacies to be observed."

So the point here is that many of the "target concepts" and "anchor words" that Fu et al. take from the NorthEuraLex database are not really translation equivalents across the 53 languages in their study, contrary to their initial assumption. This matters for their investigation of "commonalities" because, despite the initial non-equivalence of the word meanings, they still found that the neurocognitive model disclosed greater crosslinguistic similarities than the other models. And it matters for their investigation of "variation" because the initial non-equivalence already embodies some variation.

2. Another limitation of the NorthEuraLex database is that it excludes LOTS (literally thousands) of languages. MANY lexical-semantic similarities and differences have been found in studies that had more globally distributed language samples, and while some of these discoveries seem to fit Fu et al.'s climate-based account, others don't.

Here's an example that's compatible with the climate-based account. Brown (2005) found that in a global sample of 617 languages, only 389 (63%) have separate terms for arms and hands. Consistent with a previous study by Witkowski and Brown (1985) that involved a smaller sample, he also found that languages that do make this distinction tend to be located in non-equatorial zones where cold weather is frequent. Given this correlation with latitude and, by extension, temperature, he proposed that because societies in these regions are more likely to wear clothes that increase the salience of the arm-hand boundary (e.g., gloves, mittens, and long sleeves ending at the wrist), they may be more inclined to develop separate terms for arms and hands.

An example that seem to not fit the climate-based account involves spatial frames of reference. This issue is discussed in Majid et al.'s (2004) Box 1.

Another concern of mine has to do with the fMRI results. Fu et al. found that crosslinguistic semantic alignment in the neurocognitive model was associated with neural activity in bilateral ATls and left angular gyrus. But each target concept in each of the 53 languages was represented by a 300-dimensional vector, and in Fu et al.'s own previous experiment (Fu et al., 2023), vector-based representations of word meanings didn't correlate with activity patterns in ANY left perisylvian cortical areas; instead, only graph-based representations led to significant correlations. Now, other fMRI studies have in fact found significant mappings between vector-based representations of word meanings and left perisylvian cortical areas, but the details of these mappings have been inconsistent across models and studies (e.g., Carota et al., 2017, 2021, 2024; Liuzzi et al., 2023). This variability reduces confidence in Fu et al.'s new findings.

Brown, C.H. (2005). Hand and arm. In M. Haspelmath, M.S. Dryer, D. Gil, & B. Comrie (eds.), *The world atlas of language structures* (pp. 522-525). Oxford, UK: Oxford University Press.

Carota, F., Kriegeskorte, N., Nili, H., & Pulvermüller, F. (2017). Representational similarity mapping of distributional semantics in left inferior frontal, middle temporal, and motor cortex. *Cerebral Cortex*, 27, 294-309.

Carota, F., Nili, H., Pulvermüller, F., & Kriegeskorte, N. (2021). Distinct fronto-temporal substrates of distributional and taxonomic similarity among words: Evidence from RSA of BOLD signals. *NeuroImage*, 224, 117408.

Carota, F., Nili, H., Kriegeskorte, N., & Pulvermüller, F. (2024). Experientially grounded and distributional semantic vectors uncover dissociable representations of conceptual categories. *Language, Cognition, and Neuroscience*, 39, 1020-1044.

Dellert, J., Daneyko, T., Münch, A., Ladygina, A., Buch, A., Clarius, N., et al. (2020). NorthEuraLex: A wide-coverage lexical database of Northern Eurasia. *Language Resources and Evaluation*, 54, 273-301.

Fu, Z., Wang, X., Wang, X., Yang, H., Wang, J., Wei, T., Liao, X., Liu, Z., Chen, H., & Bi, Y. (2023). Different computational relations in language are captured by distinct brain systems. *Cerebral Cortex*, 33, 997-1013.

Liuzzi, A.G., Meersmans, K., Storms, G., De Deyne, S., Dupont, P., & Vandenberghe, R. (2023). Interdependency of coding for affective similarities and for word co-occurrences in temporal perisylvian neocortex. *Neurobiology of Language*, 4, 257-279.

Majid, A., Bowerman, M., Kita, S., Haun, D.B.M., & Levinson, S.C. (2004). Can language restructure cognition? The case for space. *Trends in Cognitive Sciences*, 8, 108-114.

Majid, A., Jordan, F., & Dunn, M. (2015). Semantic systems in closely related languages. *Language Sciences*, 49, 1-18.

Thompson, B., Roberts, S. G., & Lupyan, G. (2020). Cultural influences on word meanings revealed through large-scale semantic alignment. *Nature Human Behaviour*, 4(10), 1029-1038.

(Remarks on code availability)

none

Reviewer #6

(Remarks to the Author)

(Remarks on code availability)

Version 1:

Reviewer comments:

Reviewer #4

(Remarks to the Author)

We were very pleased to read a revised version of the article, and we think the paper is now in good shape. The authors took our comments seriously and implemented all the control analyses we asked for.

The biggest issue in the previous round was the inconsistency across analyses. The authors fixed that in the revision: the paper now uses a single approach across studies 1B/2/3 (language-level RSA with mixed-effects models that include climate, culture, geography, and linguistic history in the same model, plus random intercepts for language families). This makes the comparisons across studies clean and interpretable. Importantly, the results appear to be robust under this analysis: climate remains a reliable predictor of cross-lingual semantic alignment across studies.

Overall, we think that this is a strong revision. From our side, this looks publishable.

Strengths

The authors implemented several changes that improved the paper, including (but not limited to):

The analytic procedures and exact model specifications are now comparable across experiments. This was the main concern with the previous version of the manuscript and now the authors have fixed it. The authors now also show lmer formulas that make the random effects structure clear.

The link between the “neurocognitive approach” and cross-lingual similarities is now clear (ll. 108-115).

Language relatedness uses a tree-based distance (lang2vec), not a 0/1 dummy (p. 23).

The terminology is now more precise (rephrased “language models” to “word embeddings” for FastText; FastText is now described as word + subword embeddings; the descriptions of the different semantic models in p. 6 are clearer; the term “semantic projection” is now used correctly).

Weaknesses

We disagree with the authors’ interpretation of the brain results (Study 3). As we elaborated in the previous round of review, the intact vs. degraded contrast does not indicate “neural pattern similarities” (from the authors’ response), but rather, where the peak responses to language are located within the region of interest—in other words, the topographical organization of the language-selective voxels in the right ATL. The paper argues that climate modulates these topographies.

This result is surprising given that meaningful cross-lingual differences in neural activations are difficult to find on top of the substantial inter-individual variability in these patterns (especially when the numbers of data points are so low for each language). Also, we cannot fathom any sensible explanation for how climate could affect the language areas’ topography. We are therefore concerned that this result is spurious—similar to the finding that cultural (as opposed to climatic) factors influence the location of domain-general reasoning voxels (Figure S7). That said, these concerns do not, in our view, preclude publication. (If the authors did want to understand this effect a little better—at least to see whether the effect has to do with the peak location of the language region, or with some features of the topography surrounding the peak—they could e.g., remove the peak cluster from each participant’s fROI and see if the effect holds, or conversely, remove information from anything except for the peak cluster location to see if the effect holds.)

More minor issues

We disagree with the authors’ response that evaluating fastText embeddings in the native space is “challenging due to the non-isomorphic nature of embedding spaces across languages” (p. 14 of the response). This is not an issue for RSA, since it does not require direct alignment of embedding spaces. What matters is the relative structure of pairwise distances within each language’s embedding space. RSA evaluates whether the representational geometry / similarity structure is comparable, independent of absolute coordinate systems. It might be valuable (though not strictly necessary) to include an additional analysis using fastText embeddings in the native space.

In Experiment 1, the authors implement two controls: the random words model (projecting embeddings onto 100 random words) and the random dimensions model, where embeddings are projected onto 13 pseudo-dimensions. They found that the latter, with 13 dimensions, is more aligned across languages than the former. Thus, it seems that having fewer

dimensions may improve the alignment. The “neurocognitive” model, which is shown to outperform other models in alignment across languages, has fewer dimensions than the alternatives. We would recommend that the authors more directly address this issue of dimensionality. A stronger demonstration of the model’s strength would involve showing that its performance cannot be explained solely by dimensionality reduction.

Suggestions

We think the article now reads well, but there is room for improvement. A suggestion for the authors to consider is to bring climate in more prominently in the introduction and in the abstract—to us, the core contribution of the paper is showing how climate modulates semantic representations across languages, but it is only mentioned at the end of the abstract (l. 33) and the introduction.

In sum, the paper is much improved and addresses the main concerns from the first round. The results are now presented in a consistent way, and the findings on climate and semantic alignment appear to be robust. A few open questions remain, but in our view, they do not stand in the way of publication.

(Remarks on code availability)

n/a

Reviewer #5

(Remarks to the Author)

The authors adequately addressed my concerns.

(Remarks on code availability)

Reviewer #6

(Remarks to the Author)

(Remarks on code availability)

Version 2:

Reviewer comments:

Reviewer #4

(Remarks to the Author)

We were already pleased with the previous version of the article, and we confirm the paper is in very good shape. In the previous round of review, the authors had already addressed and resolved all the substantive issues with the article. Our main point of disagreement concerned the interpretation of the neural data (Experiment 3).

This latest revision:

- Clearly states that the brain results are exploratory (throughout the paper: abstract, introduction, results, discussion).
- Includes robustness checks showing that the location of the peak cluster (centroid of top 10% of the voxels) is not what is modulated by climate, and the effects remain significant controlling for that.
- Provides a conceptual clarification on their interpretation of the r-ATL findings. (We still have questions concerning this interpretation; see below).

Again, we think this paper is a strong contribution and could be published in its current form. We have, however, one comment regarding the interpretation of the findings (in particular, two sentences added in this latest revision). We believe those considerations have big implications, and we suggest either elaborating on that or removing those considerations altogether.

At p. 11, ll. 391–395, you have added the following text to clarify your interpretation of the r-ATL findings:

(a) “These neural RDMs capture the spatial pattern of activation magnitudes across voxels, reflecting both which voxels are activated and how strongly they respond. Given that different ATL subregions show preferential engagement for different semantic dimensions, variations in these distributed activation patterns may reflect differential recruitment of semantic components.”

Analogously, in the response letter, you write:

(b) “We view this pattern variation as functionally meaningful because different ATL subregions show preferential engagement for different semantic dimensions (e.g., concrete objects in ventral ATL, emotional information in medial ATL, social information in dorsolateral ATL; Visser et al., 2010; Hung et al., 2020; Wang et al., 2019). Thus, variations in the distributed activation pattern—not just peak location—could reflect differential recruitment of these semantic components.”

The t-map being used is a contrast obtained between intact and degraded speech. It tells how strongly each voxel responds to language vs. perceptually matched controls, and not in a particular sentence versus another, but across multiple sentences (diverse in semantic content). Following the considerations listed above, we believe the (semi-implicit) interpretation of the findings is:

- (1) In the r-ATL, some voxels are preferentially tuned to some semantic dimensions (say, emotional information). (We don't know the location of those putative meaning-tuned voxels in the dataset you are using.)
- (2) Climate modulates the extent to which language comprehension engages those voxels.

Our understanding is that from these points, it follows that:

- (3) Climate modulates the extent to which a given language expresses certain semantic dimensions in general—not when referring to a particular concept or when processing a particular sentence, but when processing any linguistic input.
- (4) Climate does not modulate the location of the voxels tuned to semantic dimensions. The location of those meaning-tuned voxels would be assumed to be relatively stable across languages and individuals, and what changes is the extent to which those voxels are recruited by language processing—which in turn, is determined by climate.

Point (3) is not a problem per se, but it is a very strong conclusion. If this is indeed the implication you intend to make, we'd recommend stating this more clearly.

Point (4), however, seems in tension with the rest of the manuscript. The paper's main claim is that climate modulates semantic structure across languages. However, following the interpretation from the new sentences added in (a), the semantically-tuned voxels would be assumed to be stable across individuals and languages, and what would instead vary with climate is the precise topography of language-selective voxels identified by the localizer. This inverts the logic of the paper: semantic organization becomes fixed, and the organization of the language system depends on climate.

Given this, we would recommend removing (a) altogether. As we noted before, there is currently no convincing mechanistic account of the neural finding in Study 3, and this attempt to hint at one introduces conceptual inconsistencies with other arguments laid forth in the paper.

Besides this very specific, conceptual point, we believe the paper can be published and no further review is necessary. All other issues raised in the previous round(s) have been addressed. The analyses are fully consistent across studies, the terminology is precise, the writing is clear, and the findings are interesting. This is a very nice piece of work, and we appreciate the opportunity to read it and the care in responding to our feedback.

(Remarks on code availability)

Reviewer #6

(Remarks to the Author)

(Remarks on code availability)

Response to Reviewers

We sincerely thank the reviewers for their thoughtful and constructive feedback on our manuscript. These comments and suggestions have greatly helped us improve the study. We have carefully revised the manuscript accordingly. Below, we explain how each point was addressed and the corresponding revisions.

Reviewers' comments:

Reviewer #1 (Remarks to the Author):

I have seen previous versions of this manuscript, and I approved them, as far my expertise is concerned, so I also approve this version. Data and code are much clearer than before, minor points from my side were addressed. Bigger concerns regarding the approach and conclusions that are beyond my expertise may be raised, but it is beyond my abilities to bring them up here. So, I agree with the publication of the manuscript.

Reviewer #1 (Remarks on code availability):

After previous discussions with the authors, they have tried to comply with my requirements to provide replicable code, specifically providing an extensive README file, but also program requirements, and instructions of how to run the experiments on one's own machine. I am happy with what I see here, but I have to admit I did not actively RUN the code for time constraints, as Nature Comms only gives me two weeks for the reviews and sends annoying emails to remind me to make them earlier.

Response: We sincerely thank Reviewer #1 for their continued support of our manuscript and for acknowledging the improvements made to our data and code availability. We are pleased that our README file, program requirements, and instructions meet their standards for reproducible code.

Reviewer #4 (Remarks to the Author):

Summary:

Fu et al.'s study tackles two main questions:

How well do different semantic frameworks capture cross-lingual alignment in meaning?

What variables affect inter-language differences in alignment?

To address question (1), the authors compare their primary candidate, i.e., the “neurocognitive framework” (henceforth NF, which represents meaning as loadings onto 13 cognitively relevant semantic dimensions, such as color, shape, time, space),

with a) distributional semantic models, b) semantic feature models, and c) statistical control models. They find that the NF better captures cross-lingual alignment in semantics, as shown by analyses on word embeddings in multiple languages and a colexification network.

To address question (2), the authors use representational similarity analysis, assessing the second-order similarity between “semantic” RDMs (based on word embeddings, human ratings, and fMRI data) and RDMs based on geographic, linguistic, cultural, and climatic information. The core finding is that climate-based RDMs explain cross-lingual variation in semantic structure.

Evaluation:

We find the hypothesis that climate might modulate cross-lingual differences in semantics intriguing and the motivation for investigating this question is convincing: climate strongly modulates sensory signals across modalities, and given that meaning is tied (at least in part) to sensory information, it follows that climate is a reasonable candidate for modulating semantic idiosyncrasies. We also commend the methodological breadth of the article, spanning computational modelling, behavioral experimentation, and neuroimaging; and we find the baseline conditions to be strong (with two complementary statistical control models). Nonetheless, we have serious concerns about the methodological approaches adopted in the various studies, which need to be addressed.

Response: Thank you for the positive evaluation on our study and the constructive suggestions!

The main concern

Representational similarity analyses. The most substantial concern we have is with how representational similarity analyses are conducted across the studies. In the three studies assessing the impact of climate on semantics (Study 1B, 2, and 3), the authors adopt ad-hoc statistical testing where each dependent variable is modeled with a different statistical approach:

In Study 1B, the authors predict NF-based RDMs for individual languages with RDMs based on climate, geographic, linguistic history, and culture variables. The analysis involves a crossed random-effects structure that nests language pairs within language families.

In Study 2, the authors construct RDMs for individual participants. Unlike in Study 1B, they convert the climate RDMs to ranked data. The mixed effects models involve subject pairs nested within language families (but not languages).

In Study 3, the authors use a hierarchical approach by which they compare nested models: first, a “baseline” model incorporating geographic distance and linguistic family as phylogenetic controls, and then a target model that additionally includes climate and cultural distances. The RDMs are constructed at the level of individual

participants. The random effect structure only includes random intercepts for each participant dyad. The nested models are compared by means of ΔAIC .

The three approaches are very different: sometimes the distance matrices are converted to ranked data (Study 2), the random effect structure of the mixed models varies from study to study, and while Study 1B and Study 2 simply examine the joint influence of several variables on the outcome variable, Study 3 adopts a model comparison approach. Most critically, the paper makes inferences about inter-language differences, but only Study 1B analyzes the data at the level of individual languages, whereas both Study 2 and Study 3 analyze the data at the level of individual participants.

If these studies are to be combined in a single study, the analytical approach should be consistent across them. One convincing approach would be to analyze the data at the level of individual languages (aggregating across participants) within a model-comparison framework, where a baseline model (including geographic, cultural, and linguistic history distances) is compared with a model that additionally includes climate, with random effects (ideally intercepts and slopes) for the languages in the pair. The key advantage of this analytic approach (where the analyses are conducted at the level of individual languages) is that it would most directly support the key claims of the paper about inter-language differences. Moreover, the approach where multiple participants from the same language are included in the RDMs may inflate the second-order similarity between climate RDMs and semantic RDMs simply because speakers of the same language have the same climate and possibly similar semantics. However, we acknowledge that other analytic solutions are possible; the only critical conditions are: (i) using the same analytic approach across studies and (ii) providing theoretical reasons why there might be (small) differences in the analyses across experiments. If the findings prove robust and come out reliably when the same analytic procedure is applied across studies, this would make us (and likely other readers) more confident that the findings are real and meaningful, and would likely increase the long-term impact of this work.

Response: We greatly appreciate this suggestion. Indeed, we had chosen analysis methods considering the data properties for each study separately. We fully agree with you that having a more coherent analysis scheme across studies would make the work stronger. Following this recommendation, we have updated the analyses to use a consistent analysis scheme across studies. The key findings remain robust (see Table R1 below).

We have made the following revisions to reflect this updated analysis scheme and corresponding results.

For the analytical procedures, we have added an overall paragraph explaining our unified analytical framework. In the Results section “Variations associated with environmental variables”: *“To this end, we implemented a consistent analytical*

framework across all three studies (i.e., studies with word embeddings, behavioral ratings, and neural activity data). At the language level (aggregating participants where applicable), we constructed linear mixed models that included all four variables of interest simultaneously - geographic, cultural, linguistic history, and climate variables - with random intercepts for language families to account for phylogenetic relatedness (Jackson et al., 2023; Thompson et al., 2020). This approach allowed all factors to compete within the same model, revealing their unique contributions to semantic variations. For Studies 2 and 3, we additionally conducted predictions at the individual subject level, confirming that patterns remained consistent across analytical scales.” (p.8–9). We have also substantially revised the Methods (p.19-20; p.22; p.23) to present these analytical procedures in greater detail.

For the updated results, we revised the Results sections (p.9-12). To summarize briefly here, the consistent analytical approach reveals remarkable convergence across all datasets (summarized in Table R1). Climate distance emerges as a significant predictor of semantic similarity in each dataset, with standardized beta coefficients ranging from $\beta = 0.12$ to $\beta = 0.53$, even when controlling for cultural, geographic, and linguistic history distances. The association patterns between climate and neurocognitive semantic variations were also robust across different model specifications (see Supplementary Text S3).

Table R1 [Table2 in the Revised Manuscript]. Standardized regression results for environmental predictors of semantic distance across four datasets

Data Types	Climate	Cultural	Geographic	Linguistic History
Word embedding data (wiki + CC)	0.28 (0.20, 0.37) P = 1.50×10^{-9}***	0.07 (-0.01, 0.14) P = 0.10	0.05 (-0.03, 0.13) P = 0.24	0.22 (0.16, 0.28) P = 1.29×10^{-13}***
Word embedding data (wiki + Subs)	0.44 (0.33, 0.58) P = 1.39×10^{-10}***	0.22 (0.12, 0.33) P = 4.64×10^{-5}***	-0.18 (-0.29, -0.06) P = 0.002**	0.28 (0.20, 0.35) P = 7.86×10^{-12}***
Behavioral data	0.53 (0.26, 0.80) P = 0.0006***	-0.07 (-0.36, 0.25) P = 0.64	-0.22 (-0.40, -0.04) P = 0.02	0.13 (-0.17, 0.39) P = 0.34
Neural data (R-ATL)	0.12 (0.04, 0.22) P = 0.01*	0.11 (0.008, 0.21) P = 0.047*	-0.07 (-0.15, 0.002) P = 0.06	-0.02 (-0.08, 0.04) P = 0.61

Note: Values represent standardized beta coefficients with 95% confidence intervals in parentheses. Significance levels: *** P < 0.001, ** P < 0.01, * P < 0.05.

Additional major concerns

Brain activity patterns. In Study 3, the authors correlate RDMs based on the t-maps for the intact-degraded contrast (fMRI data from Malik-Moraleda, Ayyash, et al., 2022) with RDMs based on climate, geographic, linguistic history, and culture variables. We don't think that the authors characterize this result properly. For instance, in their response to Reviewer #1 (from the earlier round), they say: "We found that in the right ATL brain region, languages associated with similar climates showed similar semantics-related neural activities". However, this is not what the t-maps for the intact-degraded contrast reflect; this contrast indicates, for each voxel, how much stronger it responds to intact vs. degraded language stimuli (this contrast is often used to identify brain areas that respond to language). This contrast does not isolate semantic processing, instead including many other aspects of language processing (e.g., phonology, morphology, syntax). What these results (if real) would indicate is that climate modulates the localization of the language areas across languages. This is a very different (and very strong!) claim. This claim is barely related to the conclusions drawn from Study 1B and Study 2, which are clearly about semantics. Moreover, the results seem highly surprising given the substantial interindividual variability in the topography of the language areas; finding meaningful interlinguistic variability on top of this interindividual variability is generally challenging, especially given that the dataset the authors draw on has only two individuals from each language.

Response: We appreciate this important comment about our interpretation of the fMRI data in Study 3. Indeed, we fully agree and are aware that the "intact-degraded" contrast in this large-scale cross-linguistic fMRI study captures multiple aspects of language processing, which include but are not restricted to, semantics. In our original submission we tried to indicate that this result by itself is suggestive rather than conclusive -- the relation to semantics was inferred from the functionality of the ATL based on literature findings and not from the current analyses. However, we agree that this is rather indirect and have thus made the following improvements for clarification, consideration of caveats, and additional analyses, inspired by this important comment.

Explicit reasoning about the semantic relationship with added analyses

We explicitly caution against characterizing the intact-degraded contrast as specifically or exclusively semantic, and further discuss how several lines of evidence convergently point to a semantic component in our findings:

1. Connection to the independently derived cross-language semantic structure: The cross-language variations of the language-task neural activity pattern in the right ATL correlated with the cross-language semantic variations independently derived in Study 1. That is, the neural variation pattern at least partly contains variance explained by the semantic variations.
2. The results of a commonality analysis: We added a new commonality analysis

(see Figure R1), demonstrating significant shared variance between climate and semantic factors in explaining the right ATL brain activation pattern variations (38.20%, 95% CI [14.75%, 48.30%]). Of course, this does not mean that climate does not affect other aspects of language processes supported by the ATL, as there were also significant unique climate effects (55.58%, 95% CI [16.83%, 85.80%]) in the commonality analysis.

3. The results of the language specificity for the climate effect: We added another new control analysis, examining neural activity during non-linguistic tasks (Spatial Working Memory and Math). Using environmental factors to predict the r-ATL's neural activity, we found that climate effects are specific to language processing and are not present during non-linguistic cognitive tasks (see Figure R2).
4. Evidence from multiple lines of research in the literature that highlights ATL as a semantic hub: Our findings were localized to the ATL, which has been established as a semantic hub by extensive literature (e.g., Lambon-Ralph et al., 2017; Patterson et al., 2007). This reasoning by itself suffers from the risk of reverse inference and is considered in light of the empirical results above.

Clarifying our interpretation of climate effects

Regarding the interpretation of the results – “climate modulates the localization of the language areas across languages.” We wish to clarify that this is not our intended interpretation:

1. Our analyses focus on neural pattern similarities within a functionally defined region (r-ATL) rather than shifts in the anatomical localization of language areas. We are examining how patterns of activation within this region vary across languages, not claiming that the region itself is relocalized based on climate.
2. The commonality analysis is particularly informative here - the shared variance between climate and semantic factors (38.20%) suggests that climate partially influences the pattern in which semantic information is processed within this region, rather than determining where language is processed in the brain.
3. Our interpretation aligns with a more conservative view: that climate may shape aspects of language processing (particularly semantic representations) within established language regions, without altering the neuroanatomical organization of language areas. This interpretation is more consistent with our findings across all three studies, which focus on climate's influence on semantic content implementation supported by a universal neuroanatomical organization.

This clarification helps bridge our neural findings with the behavioral and computational results in Studies 1 and 2, providing a more cohesive framework that focuses on how climate shapes aspects of language processing within a universal neural architecture.

Methodological strengthening

We agree with you that finding meaningful cross-linguistic differences amid

substantial individual variability presents a significant challenge, especially with only two participants per language. We carefully carried out analyses also at the individual level that take into consideration individual variations while considering cross-language patterns:

As the reviewer suggested, for the main results we report language-level results by averaging neural activity patterns within each language, allowing us to isolate meaningful interlinguistic patterns.

Additionally, we have implemented individual-level analyses that account for the hierarchical structure of our data. As detailed in Supplementary Text S3, we employed a nested random effects model:

$$Semantic_Distance_{ij} = \beta_0 + \beta_1 Climate_{ij} +$$

$$(1 | Family_i/Language_i/Subject_i) + (1 | Family_j/Language_j/Subject_j) + \epsilon_{ij}$$

This model addresses the concern about having only two individuals per language by explicitly modeling the variance components at multiple nested levels (subjects within languages within families). By implementing this three-level nested random effects structure, we account for variance at the subject, language, and language family levels.

Importantly, these individual-level analyses yielded significant climate effects (Study 2: $\beta = 0.18$, 95% CI [0.17, 0.18], $P < 1 \times 10^{-16}$; Study 3: $\beta = 0.07$, 95% CI [0.03, 0.11], $P = 0.0003$), confirming that our findings remain robust despite the challenges of inter-individual variability highlighted by the reviewer.

Manuscript revision

We have made the following revision accordingly:

1. Reporting the semantic-neural associations in the r-ATL: *“We then used linear mixed regression models to assess the relationships between these neural RDMs and the 13-dimensional neurocognitive semantic model RDMs from Study 1, while including random intercepts for language families to account for the phylogenetic structure. Significant beta coefficients were found only in the right anterior temporal lobe (ATL; $\beta = 0.44$, 95% CI = [0.17, 0.71], FDR-corrected $q = 0.016$). That is, the more closely aligned two languages are for the 13-neurocognitive dimensional semantic representation, the more similar their speakers’ brain activities are in the r-ATL when processing language.”* (p.11)
2. A newly added section to characterize the effects using control analysis and commonality analysis: *“Importantly, when using environmental RDMs to predict the r-ATL’s neural activity during non-linguistic tasks (e.g., Spatial Working Memory -- Hard vs Easy; Math -- Hard vs Easy), the climate effects were no longer significant, suggesting that these effects are unique to language-related processing (Supplementary Text S5; Figure S7). To further characterize the relationship between semantic and climate effects in the r-ATL, we conducted a commonality analysis to partition the r-ATL variance explained by different components (Figure 4c, right panel). This analysis revealed significant unique contributions from both semantic factors (6.26%,*

95% CI [1%, 37.46%]) and climate factors (55.58%, 95% CI [16.83%, 85.80%]), as well as substantial common variance shared between climate and semantic effects (38.20%, 95% CI [14.75%, 48.30%]). The significant common variance suggests that climate effects on neural activities in the r-ATL partially reflect semantically related neural activation patterns.” (p.11)

3. Revised interpretation in the Discussion and relationships with Studies 1 and 2: *“This broad dimensional effect is also in line with our intriguing observation of climate-brain activity associations in the r-ATL in the human brain. Note that we analyzed brain activity for the “intact vs. degraded” contrast, which captures broad linguistic processes including, but not limited to, semantics. Regarding the ATLs’ role in language processing, accumulating research has established them as key regions for processing higher-order semantics, binding the distributed semantic dimensional representations and word relational structures in language (see reviews in Bi, 2021; Lambon-Ralph et al., 2017; Miyashita, 2019). Indeed, the commonality analyses revealed significant shared components between climate and semantics, linking ecological variables and language neural processing that converges with our behavioral and computational findings. We do not exclude other potential language processes that may also mediate the climate-ATL association -- our commonality analysis indeed also revealed significant variance shared by climate and neural activities that is not shared with semantics.”* (p.15).
4. Further consideration of caveats in the Discussion. We further specified in the Discussion that: *“Future studies should employ tasks specifically designed to isolate semantic processing across diverse languages to more precisely characterize climate-semantic neural relationships.”* (p.16).

Figure R1 [Figure 4 in the Revised Manuscript]. Neural correlates of environmental effects on language processing (Study 3). a. Methodology for constructing neural RDMs from 86 subjects across 45 languages within the high-level language processing network (Fedorenko et al., 2010). Inter-language neural RDMs were derived from the contrast between intact and degraded native language processing (Malik-Moraleda et al., 2022), with data averaged for each language. b. Linear mixed regression analysis at the language level, using semantic relation patterns (from the 13-neurocognitive dimensional space in Study 1) to predict cross-language neural variations. The bar plot depicts regression coefficients across all 12 brain regions. The right ATL is the only region showing significant beta values after multiple corrections (FDR-corrected $p < 0.05$), indicating that this region encodes cross-language semantic alignments in the 13-dimensional space derived in Study 1. c. Climate effects on neural activity patterns in the right ATL. Environmental factor RDMs were obtained for participants' native languages. The regression coefficient plot (left panel) illustrates significant climate effects in the right ATL. The commonality analysis (right panel) quantifies the proportion of variance explained by different components: unique semantic effects (6.26%, 95%CI [1%, 37.46%]), unique climate effects (55.58%, 95% CI [16.83%, 85.80%]) and common variance shared between climate and semantic factors (38.20%, 95% CI [14.75%, 48.30%]). The significant common variance indicates that climate effects on neural activities in the right ATL partially reflect semantically related neural activation patterns. Abbreviations: ATL, anterior temporal lobe; IFG, inferior frontal gyrus; IFGorb, inferior orbital frontal gyrus; MFG, middle frontal gyrus; MTG, middle temporal gyrus;

AG, angular gyrus. Error bars represent 95% CIs. Significance levels: * $P < 0.05$.

Figure R2 [Figure S7 in the Revised Manuscript]. Beta coefficients from linear mixed-effects regression models predicting neural pattern dissimilarities in the right anterior temporal lobe (r-ATL) during nonlinguistic tasks. (a) Spatial Working Memory task (Hard vs Easy condition), showing significant effects only for cultural distance ($\beta = 0.13$, 95% CI [0.02, 0.24], $P = 0.016$) but not for climate distance ($\beta = 0.06$, 95% CI [-0.05, 0.16], $P = 0.213$). (b) Math task (Hard vs Easy condition), showing no significant effects for any environmental variable (all P s > 0.05). Error bars represent 95% confidence intervals; * indicates $P < 0.05$.

Representational similarities and differences within the NF. The NF in Study 1 is based on projecting word embeddings onto a lower-dimensional space of 13 dimensions/features. The authors found that this lower-dimensional space better reflects semantic similarities across languages. This makes sense, as this space consists of a possibly universal set of conceptual dimensions and filters away variance along other potentially relevant semantic axes that might be more idiosyncratic and language-specific. Yet, they use this space to study cross-lingual differences and how those are modulated by climate. There appears to be a contradiction here—if the authors want to model cross-lingual differences in semantics, why choose a representational projection that minimizes those differences? One motivation for restricting the feature space to those semantic dimensions could be that, in principle, climate could be expected to modulate semantic differences tied to the sensory information conveyed by words, and for this reason, it would make sense to restrict the focus to those features. If this is the case, this line of reasoning should be clearly formulated in the article (see also the following point “Sensorimotor and cognitive dimensions”), because the original line of reasoning put forward by the authors (i.e., adopting the NF because it better identifies cross-lingual consistencies in semantics) is not convincing. More generally, the authors should emphasize the point we made above, namely that the NF restricts semantic variation by projecting meaning onto a small set of semantic dimensions. Relatedly, it would be interesting to see how much information is lost in this projection (partialling out information beyond the 13 NF

axes); this could be achieved by doing RSA over the native word embedding space and the NF space in the various languages.

Response: We greatly appreciate this comment! Indeed, our writing invited this confusion—after showing that the representation based on the neurocognitive framework (NF) shows less variance in the commonality analyses in Study 1, it was not optimal to study variation if the goal was to model variation maximally.

In fact, our rationale here is not to exhaustively model all cross-lingual differences in semantics (it's not clear how this would be done, see next paragraph). Rather, the goal is to understand the mechanisms underlying variations along the NF -- if this structure is neurobiologically universal, how does it adapt to different ecological environments? That is, the motivation for restricting the focus to the NF is to understand variations meaningful to these dimensions. We have made careful revisions to make this motivation explicit.

Furthermore, it is not clear how to model the “maximum” semantic variations. The reviewer suggested an interesting approach that we also considered initially -- doing RSA over the native word embedding spaces and the NF space in the various languages. But from a methodological perspective, cross-linguistic semantic alignments require common reference systems. Native word embedding spaces are trained independently within each language, creating non-isomorphic spaces that lack shared reference points (as discussed in Artetxe et al., 2018; Mikolov et al., 2013; Thompson et al., 2020). This is why our approach, like related work, relies on anchor word methodologies.

The following specific revisions were made to more clearly explain our rationale:

1. Emphasized that NF frameworks intrinsically accommodate variations when first introducing the NF model: *“We propose that the intrinsic way in which the human brain represents semantic knowledge offers a strong candidate framework for understanding both cross-linguistic semantic universality and variation. The biological constraints of the human brain are the result of the biological evolution of homo sapiens and lay the foundation for universality (see similar arguments for color space in Berlin & Kay, 1991; emotion space in Jackson et al., 2019), and such a biological structure would respond to different environmental inputs (e.g., naturally and culturally varied), resulting in phenomenal variations.”* (p.3-4)
2. Added methodological justification for using the NF framework to study differences: *“This neurocognitive approach also offers methodological advantages for cross-linguistic comparison. By focusing on dimensions with established neural correlates, we create a controlled space where both universal patterns and meaningful variations can be understood against a consistent 'backbone' of biological universality. Such dimensional structures not only reveal fundamental commonalities in semantic organization across languages due to our shared neural architecture, but also filter out potentially*

idiosyncratic semantic dimensions to highlight variations specifically along sensorimotor and core-cognitive channels." (p.4)

3. Highlighted an explicit prediction about non-linguistic variations within the framework: *"The neurocognitive framework of semantic representation intrinsically makes the following predictions regarding commonalities and variations (Figure 1)...Second, regarding variations along this dimensional structure, the intrinsic assumption of the framework that semantic representation derives from sensory channels predicts that variability is the natural consequence of variables affecting these channel-mediated experiences, including those associated with both natural and cultural environments (Kemmerer, 2023)."* (p.4-5)

References:

- Artetxe, M., Labaka, G., & Agirre, E. (2018). A robust self-learning method for fully unsupervised cross-lingual mappings of word embeddings. *arXiv preprint arXiv:1805.06297*.
- Mikolov, T., Le, Q. V., & Sutskever, I. (2013). Exploiting similarities among languages for machine translation. *arXiv preprint arXiv:1309.4168*.
- Thompson, B., Roberts, S. G., & Lupyan, G. (2020). Cultural influences on word meanings revealed through large-scale semantic alignment. *Nature Human Behaviour*, 4(10), 1029-1038.

Characterization of the alternative models. The authors present three alternatives to the NF: a) distributional semantic models, b) feature-based psycholinguistic models, and c) statistical control models. However, we believe that the authors' operationalization of feature-based psycholinguistic models (calculating the cosine similarity between a target word and the 100 most frequent English features in the Buchanan et al. (2019) norms) mischaracterizes how feature models are intended to be used (where each word is represented as a set of discrete features that characterize a concept, like "is red", "is mammal", etc.). After all, all the models they consider in their first study are distributional semantic models, where word embeddings are projected onto lower-dimensional spaces which consist of neighbouring words (a), frequent features (b), or randomly sampled words (c). Even the NF consists of a projection of word embeddings onto the 13 cognitively relevant features. Since all the considered models are ultimately projections of a distributional semantic model (DSM), we believe it is warranted to include the "native" representations of the DSM itself. This would require:

Evaluating how similar the different languages are in the native fastText word embeddings space (without projecting embeddings onto anchor words), and
Evaluating the similarity between the native fastText-based RDM and the RDMs based on climate, geographic, linguistic history, and culture.

Response: Thank you for this very helpful suggestion. Indeed, the original writing

created confusion and your suggested changes would be much clearer.

We have thus revised the corresponding section accordingly (Constructing different model-anchor-based semantic representations, p.6): *“We performed cross-linguistic comparisons of semantic representations obtained based on different semantic models. All semantic representations were obtained using an anchor word approach (i.e., essentially all are distributional semantic methods), but the key difference is what types of words were used as anchors for representations. Different anchor words were selected based on different model principles:”*

For the feature-based approach specifically, we now elaborate: *“The psycholinguistic tradition employs semantic feature models based on human-generated decompositional features (e.g., 'has fur', 'is round'), representing concepts as binary or weighted vectors indicating the presence/absence or strength of each feature (Buchanan et al., 2019; McRae et al., 2005). To adapt this approach for cross-linguistic comparison, we represent the 1,016 concepts using their embedding distance patterns with the 100 most frequent feature words from Buchanan et al. (2019).” (p.6)*

Regarding your suggestion to evaluate native fastText representations without projection, direct comparison in the original embedding space is challenging due to the non-isomorphic nature of embedding spaces across languages (see also the response to the previous comment above). However, we fully agree that considering variation patterns along other semantic structures is also valuable.

To this end, we have added analyses testing whether climate effects on cross-linguistic semantic alignment are specific to neurocognitive semantic models. As shown in Table R2 (Table 1 in the revised manuscript, p.39), climate maintains a significant association with neurocognitive semantic structure even when controlling for distributional global ($\beta = 0.03$, $p = 0.02$), distributional local ($\beta = 0.11$, $p < 0.001$), and feature-based ($\beta = 0.13$, $p < 0.001$) models. Conversely, when predicting these alternative semantic structures while controlling for neurocognitive structure, climate showed either weaker or non-significant associations (distributional global: $\beta = 0.03$, $p = 0.005$; distributional local: $\beta = -0.02$, $p = 0.37$; semantic feature: $\beta = -0.03$, $p = 0.15$).

These findings suggest a specific relationship between climate and neurocognitive semantic organization that cannot be fully explained by other semantic representation frameworks. These analyses provide additional support for our central claim that neurocognitive semantic structure captures meaningful cross-linguistic variation associated with non-linguistic environmental factors like climate. We sincerely thank the reviewer for these suggestions to strengthen our findings.

Table R2 [Table 1 in the Revised Manuscript]: Unique effects of climate on different semantic structures

Model Type	Climate Effect (β)	95% Confidential Intervals
Model 1: Neurocognitive as DV		
Neurocognitive ~ Climate + Global Dist.	0.03*	[0.01, 0.05]
Neurocognitive ~ Climate + Local Dist.	0.11***	[0.09, 0.13]
Neurocognitive ~ Climate + Feature	0.13***	[0.10, 0.14]
Model 2: Alternative Models as DV		
Global Dist. ~ Climate + Neurocognitive	0.03***	[0.02, 0.06]
Local Dist. ~ Climate + Neurocognitive	-0.02	[-0.03, 0.01]
Feature ~ Climate + Neurocognitive	-0.03	[-0.04, 0.00]

Note: *** P < 0.001, ** P < 0.01, * P < 0.05; DV, dependence variable; Global Dist., global distributed model; Local Dist., local distributed model; Feature, semantic feature model; All models control for random effects of language families; All variables were standardized.

Sensorimotor and cognitive dimensions. The NF uses 13 dimensions; 7 are sensorimotor (color, shape, ...), while the remaining 6 are called “cognitive domains” (time, space, number, ...). While it is rather clear why climate should have an effect on sensorimotor representations, the authors need to elaborate why climate would affect the “cognitive domains” dimensions as well.

Response: Indeed, the mechanisms of climate’s effect on the core-cognitive domains were not as transparent as its effects on the sensorimotor dimensions. We have now made this more explicit and added discussions: *“Notably, our results from both word embedding and behavioral rating experiments demonstrate that climate exerts a robust effect across all semantic dimensions, encompassing not only sensorimotor properties but also higher-order domains such as mental-cognition, time, space, number, social, and emotion (Figure 3c and 3f). Sensorimotor dimensions are expected to be directly shaped by environmental inputs. While higher-order cognitive dimensions are further abstracted from immediate perceptual input, they arguably remain partially grounded in sensorimotor experience (Sigismondi et al., 2024). These findings are consistent with prior evidence that environmental conditions can modulate temporal perception (Li et al., 2022), and that climate-linked subsistence patterns shape social behaviors and emotional tendencies (Talhelm et al., 2014; Van de Vliert, 2013), and they clearly invite further investigations for the underlying neurocognitive mechanisms” (p.14-15)*

Minor points:

Semantic projection. The authors say they employ the “semantic projection” method to project word embeddings onto the 13 dimensions of the NF. Nonetheless, their approach only consists in calculating the cosine between each word and a few anchor words for each dimensions, which does not correspond to the semantic projection method proposed by Grand et al. (2022)—in fact, it was found to perform worse than the actual method, which involves calculating a dimension vector as the difference between two “poles” (for the sound dimension they could be something on the lines of “noisy” vs. “silent”), and then projecting individual words onto these dimensions with the dot product. We understand that it might be difficult to define the poles for some features which are not intrinsically unidimensional (e.g., shape), but this should then be clarified.

Response: Thank you! We have revised our terminology to accurately reflect our methodology: We now use the term “anchor word approach” rather than “semantic projection” throughout the manuscript to accurately describe our method and avoid confusion with Grand et al. (2022)'s semantic projection technique.

Language and thought. From the abstract: “languages are used worldwide as the

primary means of human thought”. This is a rather controversial statement, we’d recommend the authors to tone it down in light of evidence that language and thought appear to be largely dissociable (see Fedorenko, Piantadosi, Gibson, 2024 for a recent review).

Response: Thank you. We have toned down to languages being “the primary means of human thought communications”.

Language family distance. The language family distance metric (1 if two languages belong to the same family, 0 otherwise) is a coarse metric. The authors could consider using a distance metric based on a language tree. For instance, the genetic distance calculated by the Python package lang2vec calculates the distance between two languages by counting the number of upward steps required on the tree until both languages converge at a common node (Littell et al., 2017).

Response: Thank you! Following this helpful suggestion, in Study 3 we now use this more fine-grained language family distance metric based on a language tree using the genetic distance calculated by the Python package lang2vec (see p.23). Results remained robust.

Typos and writing suggestions:

The core hypothesis of the paper (that climate might modulate cross-lingual differences in semantics because of general perceptual differences in different climatic environments) becomes clear only at ll. 130-137; we recommend to clearly specify this point earlier in the introduction so that readers can understand from the get-go what the article is about.

Response: Thank you. We now made this point earlier in the introduction: “*We propose that the intrinsic way in which the human brain represents semantic knowledge offers a strong candidate framework for understanding both cross-linguistic semantic universality and variation. The biological constraints of the human brain are the result of the biological evolution of homo sapiens and lay the foundation for universality (see similar arguments for color space in Berlin & Kay, 1991; emotion space in Jackson et al., 2019), and such a biological structure would respond to different environmental inputs (e.g., naturally and culturally varied), resulting in phenomenal variations.*” (p.3)

Throughout the article, the authors consistently refer to word embedding models as “language models”; consider rephrasing as word representation models or word embedding models. Also, fastText (the word embeddings model used in the study) is a peculiar word representation model in that it represents words as a sum of word embeddings plus subword (n-gram) embeddings. This should be mentioned somewhere in the paper.

Response: Thank you. We have made the following revisions accordingly:

1. We have replaced all instances of “language models” with more precise terminology such as “word representation models” or “word embedding models” throughout the paper.
2. We have explicitly described the unique characteristics of fastText, noting its subword mechanism in Method section: *“FastText represents words as a combination of whole-word and subword (character n-gram) embeddings, which allows for better handling of morphologically rich languages and out-of-vocabulary words (Bojanowski et al., 2017).” (p.17)*

The theoretical basis of the NF is not very clear: it starts by positing that language and sensory experience form the basis of conceptual representations (II. 77-99) and then concludes that an appropriate way to represent semantics is by using 13 dimensions (sensorimotor dimensions and “cognitive domains”), without mentioning how language information fits within this model and where the dimensions from the “cognitive domains” come from.

Response: We appreciate the reviewer's thoughtful comment about the theoretical foundation of the NF. We have clarified this in our revised manuscript.

The theoretical basis of our NF is grounded in converging evidence from cognitive neuroscience showing that semantic representation in the brain derives from both sensory (and language) experiences through specific neural processing architectures. As we explain in our manuscript, *“Neurocognitive research reveals a consensus framework that semantic representation in the brain is derived from sensory (and language) experiences in ways that respect the specific information processing architectures of the brain: Brain responses to word meanings are distributed along sensorimotor and related associative cortical networks, respecting domains of evolutionary saliency, with activity strength modulated by the meaning’s loading on corresponding attributes/domains (Fernandino et al., 2016; Fernandino et al., 2022; He et al., 2013; Martin et al., 1995).” (p.4)*

I. 63: “beauty in English, bella in Italian” – it should either be “beautiful in English, bella in Italian” or “beauty in English, bellezza in Italian”.

Response: Thank you! Corrected (see p.3: *beautiful* in English and *bella* in Italian).

II. 119-120: “These alternative models [distributional semantic models, feature models] are prevalent in cognitive disciplines without explicit considerations of the neurobiological basis and have been shown to be less correlated with word semantic brain activities than the neurocognitive dimensional model [...]” – distributional semantic models and their successors (e.g., language models) are often linked to neurobiological principles, and obtain great fit to brain activity patterns (see e.g.,

Goldstein et al., 2022; Shrimpf et al. 2021).

Response: Agreed. We now removed the statement which might introduce controversial and more accurately characterize models as *“other distribution-based models (Figure 2a) that do not explicitly incorporate the neural architecture...”* (p.4)

I. 299: It would be great if the authors could write the lmer formulas to better understand the random effect structure of the mixed models (also, I. 347).

Response: Thank you! Following the recommendation, we have now added the lmer formulas in the main texts (p.19; p.22; p.23).

I. 352: The p-value is missing for the coefficients of “linguistic history”.

Response: Added.

II. 211-212: In the context of the finding that the “global” distributional space finds more cross-lingual commonalities than the “local” distributional space, consider citing Lewis et al. (2023), who found the opposite pattern.

Response: Thank you for the valuable suggestion to reference Lewis et al. (2023). We have added an explanation about the seemingly different findings (due to different methods) and implications.

In our revised discussion, we explain that our definition of “global” differs from Lewis et al. (2023): Our “global” includes all word relationships in the vocabulary space (i.e., *grand global*), while Lewis et al. specifically separated cross-domain (“global”) from within-domain (“local”) word relations, i.e., excluding near neighbor relations from their global computations. The finding that including all words as anchors (in our results) shows greater cross-linguistic commonality, while including only distant words as anchors does not (Lewis et al., 2023), indicates that preserving domain structure is important in capturing cross-linguistic semantic universals.

We have incorporated this point into our Discussion section (p.13-14): *“Note that compared to Lewis et al (2023), which also studied distributional models and showed that global distribution space (excluding within-domain neighbors as anchors) showed less universality compared to local space (within-domain neighbors as anchors), our analyses of global distribution space included all word relationships in the vocabulary space (i.e., grand global), and showed greater universality. These results, when considered together, indicate that preserving domain structure is important in capturing cross-linguistic semantic universals.”*

I. 596: “billions to hundreds of millions” → “hundreds of millions to billions”

Response: Revised.

I. 713: “with Fisher-Z transformed” → “with Fisher’s r-to-z transformation”

Response: Revised.

Questions:

Distributional local vs. global models. It is unclear how the distributional local models work. The authors say that in the local condition, each word is projected onto its immediate neighbors. We do not understand how this might work—if each word is projected onto its neighbors and the neighbors change from word to word, this would create a dimensionality mismatch (e.g., w_1 would be projected onto w_2, w_3, w_4 etc., while say w_{123} would be projected onto w_{124}, w_{125} etc., so the dimensions of the projection of w_1 would be different from the dimensions of the projection of w_{123}). Could you please clarify how this approach works? Relatedly, the global models use as anchors all the words but the target word itself. This would once again cause a dimensionality mismatch, as, for instance, the anchor w_1 would be absent from the embedding projection of w_1 , and the anchor w_{123} would be absent from the embedding projection of w_{123} .

Response: Thank you for this important clarification question about our distributional models. We indeed maintained consistency in cross-linguistic comparisons by using the same set of anchor concepts (all words other than the target word) across languages.

The cross-language similarity calculation was performed first for each word. Using your example, in global models for a given language pair, the embedding projection of W_x would use all words except for W_x , i.e., the same anchor words for each target word, yielding a correlation value for W_x for that language pair. The following analyses were computed from these correlation values (e.g., averaging the correlation coefficients of all target words to obtain the correlation coefficient for the language pair).

In the local distributional models, we chose the shared neighbors across each language pair. Specifically, for any given word (e.g., “dog”), we identified the nearest semantic neighbors ($K=100$) in the embedding space in each language (e.g., English “dog” vs. Chinese “狗”), then took the intersection of these neighbor sets across each language pair. This ensures that we only used anchors that had correspondences in both languages, maintaining consistent dimensionality in the cross-linguistic comparisons.

References

Lewis, M., Cahill, A., Madnani, N., & Evans, J. (2023). Local similarity and global

variability characterize the semantic space of human languages. *Proceedings of the National Academy of Sciences*, 120(51), e2300986120.

Littell, P., Mortensen, D. R., Lin, K., Kairis, K., Turner, C., & Levin, L. (2017). Uriel and lang2vec: Representing languages as typological, geographical, and phylogenetic vectors. *Proceedings of the 15th Conference of the European Chapter of the Association for Computational Linguistics: Volume 2, Short Papers*, 8–14.

Fedorenko, E., Piantadosi, S. T., & Gibson, E. A. (2024). Language is primarily a tool for communication rather than thought. *Nature*, 630(8017), 575-586.

Reviewer #5 (Remarks to the Author):

Fu et al. address an important yet neglected topic--namely, crosslinguistic similarities and differences in meaning, and their implications for cognition--by combining computational linguistics and neuroscience. Their methods are innovative and their results are interesting and valuable. I do, however, have some concerns.

Response: We greatly appreciate the positive evaluations of our study.

Like the study by Thompson et al. (2020), this one relies heavily on Dellert et al.'s (2020) NorthEuralex database, but this database has several limitations that Fu et al. should explicitly acknowledge. Two major limitations are as follows.

1. Fu et al. state that the “target concepts” and “anchor terms” they draw from this database are “translation equivalents” across the 53 languages they focus on. But this is not true. Even Dellert et al. are quite open about this. They state: “For instance, many languages lexically distinguish the substance one breathes from the space in which birds move, making it necessary to further specify the English equivalent “air”. For some concepts (especially kinship terms), the specification also defines which term is to be mentioned first in our list of equivalents if a more fine-grained distinction is made in the target language, e.g., ‘grandfather (e.g., father’s father)’.”

All of the information in the database was compiled from dictionaries, without extensive validation from native speakers. This is another shortcoming that Dellert et al. (2020) admit and that Fu et al. should make clear too.

Crosslinguistic lexical-semantic studies that HAVE included native speaker judgments have revealed nontrivial differences in how even closely related languages partition the same conceptual domains, and this has implications for Fu et al.'s study. As described below, two examples involve color terms and body part terms, but comparable results have been reported for many other domains.

For instance, Fu et al. use lots of basic color terms (e.g., blue, red, yellow, etc.) as “anchors” for their neurocognitive color dimension. But Majid et al. (2015) show that there are not straightforward translation equivalents for these terms across 12 Germanic languages. Here's a snippet of their discussion: “Within the colour domain, English and Danish differed most from one another (see Table 2 with correlations).

English red is very localized: it was the dominant response for only one colour chip. In contrast, Danish rød was a dominant response for 11 different chips. In English, speakers preferentially called most of those chips pink, which was then also extended into an area that Danish speakers called lilla. Similarly, English orange was smaller in extension than its Danish equivalent orange. English yellow and Danish gul are approximately the same size, but because Danish orange has a larger extension gul seeps into territory that English speaker preferentially name green.”

The body part domain is relevant to Fu et al.'s study because they used lots of body part terms as “anchors” for their neurocognitive motor dimension. But Majid et al. (2015) report nontrivial lexical-semantic differences between 12 Germanic languages for such terms. Here's part of their discussion: “...there is no single cognate set used in the Germanic languages for ‘leg’, and we see a more divergent pattern of naming. Participants saw red dots on the upper and lower leg, on the right and left side of the body, viewed from front and back. This lets us examine how they chose to refer to these distinct images. English speakers predominantly used leg for all such stimuli; similarly Dutch, Flemish and Luxembourgish used a single term been/Been. Frisian speakers also used been, but not as frequently. However, the speakers of the other languages tended to use distinct terms for the upper and lower leg. German speakers used Oberschenkel and Unterschenkel, literally ‘upper-leg’ and ‘lower-leg’, but the root *Schenkel alone was never attested. Schwyzerdütsch speakers used the cognate terms, obèrschènku and ungerschènku, although bare schènku was also used (but only twice, and only with reference to upper leg stimuli). The Scandinavian languages make sporadic use of bein/ben, but for the most part other terms were more frequent. Like German speakers, the participants preferentially distinguished upper and lower leg. This discussion illustrates that even in a domain with much shared structure, there are still many intricacies to be observed.”

So the point here is that many of the “target concepts” and “anchor words” that Fu et al. take from the NorthEuraLex database are not really translation equivalents across the 53 languages in their study, contrary to their initial assumption. This matters for their investigation of “commonalities” because, despite the initial non-equivalence of the word meanings, they still found that the neurocognitive model disclosed greater crosslinguistic similarities than the other models. And it matters for their investigation of “variation” because the initial non-equivalence already embodies some variation.

Response: We greatly appreciate this comment. We fully agree that the proxy term “translation equivalent” is misleading: methodologically the imperfect translations in NorthEuraLex should be noted; theoretically, the whole point of the study is exactly to address cross-linguistic similarity/differences. As presented in our Introduction, we explicitly acknowledge the cross-linguistic lexical-semantic differences highlighted by the reviewer: *“Ethnographic descriptions have identified common concepts across semantic domains such as color (Berlin & Kay, 1991) and emotion (Ekman, 1992; Wierzbicka, 1992), while also revealing different words for the same perceptual referents, culturally bound words resistant to translation, and different categorization systems (Majid et al., 2004; Majid et al., 2015; Majid et al., 2018; Passmore & Jordan,*

2020).” (p.3)

We have deleted the term “translation equivalent” to avoid confusion, which did not affect the data description. The related critical issue is whether the empirical findings and implications are affected by the fact that the target words tested and the anchor words we chose are not fully translation equivalent across languages.

Regarding commonalities, the key comparison was the magnitude of the commonality across different models, using the same set of concept translations. That is, the robust cross-linguistic convergence in neurocognitive dimensions, relative to other models, was not attributable to word selection choices.

Regarding variations, the reviewer's concern that “the initial non-equivalence already embodies some variation” is particularly insightful. To address this directly, we conducted new analyses to determine whether the climate-related effects we observed might be artifacts of lexical-semantic differences inherent in the database. If our climate findings were merely reflecting these pre-existing lexical differences (which would still be interesting), then climate should show similar associations with all semantic models, since all models use the same imperfect translation equivalents. However, our new analyses (Table R2; Table 1 in the revised manuscript, p.39) show that climate maintains a significant association with neurocognitive semantic structure even when controlling for other semantic models (distributional global: $\beta = 0.03$, $p = 0.02$; distributional local: $\beta = 0.11$, $p < 0.001$; feature-based: $\beta = 0.13$, $p < 0.001$). Conversely, when predicting alternative semantic structures while controlling for neurocognitive structure, climate showed either weaker or non-significant associations (distributed global: $\beta = 0.03$, $p = 0.005$; distributed local: $\beta = -0.02$, $p = 0.37$; semantic feature: $\beta = -0.03$, $p = 0.15$). This pattern suggests that the climate effects we observe are specifically related to neurocognitive semantic dimensions rather than being artifacts of the imperfect translation equivalents in the NorthEuralex dataset.

Furthermore, our complementary Studies 2 and 3, which use different methodologies that do not rely on the anchor word approach, yield similar climate-related effects, providing converging evidence that our findings reflect cognitively related semantic patterns rather than methodological artifacts from the NorthEuralex database.

In the Revision, we have added these additional validation analyses (p. 9). We have also added a specific caveat consideration section to bring attention to this issue: *“Second, true translation equivalence across languages is impalpable given theoretical and empirical considerations. We tested words from the NorthEuralex database, which contains dictionary-compiled translations. As Dellert et al. (2020) noted, many languages make lexical distinctions absent in English (e.g., differentiating “air” as a breathable substance versus a space for flight), and these mappings lack extensive native speaker validation. Despite these inherent limitations, our study focuses on the*

underlying principles governing such cross-linguistic (mis)alignments.” (p.15).

Thank you for prompting us to strengthen the paper with these important considerations.

Table R2 [Table 1 in the Revised Manuscript]: Unique effects of climate on different semantic structures

Model Type	Climate Effect (β)	95% Confidential Intervals
Model 1: Neurocognitive as DV		
Neurocognitive ~ Climate + Global Dist.	0.03*	[0.01, 0.05]
Neurocognitive ~ Climate + Local Dist.	0.11***	[0.09, 0.13]
Neurocognitive ~ Climate + Feature	0.13***	[0.10, 0.14]
Model 2: Alternative Models as DV		
Global Dist. ~ Climate + Neurocognitive	0.03***	[0.02, 0.06]
Local Dist. ~ Climate + Neurocognitive	-0.02	[-0.03, 0.01]
Feature ~ Climate + Neurocognitive	-0.03	[-0.04, 0.00]

Note: *** P < 0.001, ** P < 0.01, * P < 0.05; DV, dependence variable; Global Dist., global distributed model; Local Dist., local distributed model; Feature, semantic feature model; All models control for random effects of language families; All variables were standardized.

2. Another limitation of the NorthEuraLex database is that it excludes LOTS (literally thousands) of languages. MANY lexical-semantic similarities and differences have been found in studies that had more globally distributed language samples, and while some of these discoveries seem to fit Fu et al.'s climate-based account, others don't. Here's an example that's compatible with the climate-based account. Brown (2005) found that in a global sample of 617 languages, only 389 (63%) have separate terms for arms and hands. Consistent with a previous study by Witkowski and Brown (1985) that involved a smaller sample, he also found that languages that do make this distinction tend to be located in non-equatorial zones where cold weather is frequent. Given this correlation with latitude and, by extension, temperature, he proposed that because societies in these regions are more likely to wear clothes that increase the salience of the arm-hand boundary (e.g., gloves, mittens, and long sleeves ending at the wrist), they may be more inclined to develop separate terms for arms and hands. An example that seem to not fit the climate-based account involves spatial frames of reference. This issue is discussed in Majid et al.'s (2004) Box 1.

Response: We thank the reviewer for raising this important point about the geographical limitations of the NorthEuralex database and for suggesting relevant research from more globally distributed language samples.

We fully acknowledge that the NorthEuralex database excludes thousands of languages, with a clear Eurasian bias that limits the global representativeness of our primary analyses. We have now addressed this limitation in our revised manuscript, incorporating the very helpful points noted by the reviewer:

1. We expanded our analysis beyond the 53 languages in NorthEuralex in the "Generalization to colexification networks with 2,681 languages" section (p.7), where we generalize the examined patterns to a more diverse set of languages.
2. In our revised Discussion, we've added a specific section addressing how a more globally distributed language sample might affect our findings: *"...while our language samples included in the environment-semantic association analyses cover a substantial portion (approximately 1/2 - 1/3) of the world's major populations across the three studies, they have all undergone varying degrees of modernization. This may lead to an overestimation of the climate effects and an underestimation of the effects of cultural and linguistic factors on semantic structures, which warrants further investigation. Future research with more globally representative language samples is necessary to extend the generalizability of our findings."* (p.16)

We have also incorporated both types of examples provided by the reviewer in our revised Discussion. While our study aims to explore similarities and variations at the structural level, we now point out that whether climate influences within-domain variations remains unresolved and that future studies are needed: *"However, evidence regarding climate's influence on within-domain semantic variations shows mixed patterns - Brown (2005)'s findings on arm-hand lexical distinctions in colder regions*

support climate-based accounts, while spatial reference frame variations (absolute vs. relative) may be more influenced by a complex set of factors rather than by a single factor such as climate (Majid et al., 2004). More generally, having identified the robust association between climate and a central cognitive component — semantics, future studies elucidating the detailed neurocognitive mechanisms would be critical for understanding the intricate interplay between our living environment and diverse human behaviors, especially in the era of profound climate change.” (p.15)

Another concern of mine has to do with the fMRI results. Fu et al. found that crosslinguistic semantic alignment in the neurocognitive model was associated with neural activity in bilateral ATLs and left angular gyrus. But each target concept in each of the 53 languages was represented by a 300-dimensional vector, and in Fu et al.'s own previous experiment (Fu et al., 2023), vector-based representations of word meanings didn't correlate with activity patterns in ANY left perisylvian cortical areas; instead, only graph-based representations led to significant correlations. Now, other fMRI studies have in fact found significant mappings between vector-based representations of word meanings and left perisylvian cortical areas, but the details of these mappings have been inconsistent across models and studies (e.g., Carota et al., 2017, 2021, 2024; Liuzzi et al., 2023). This variability reduces confidence in Fu et al.'s new findings.

Response: We appreciate the reviewer for highlighting the seemingly inconsistency between our current findings and our previous work (Fu et al., 2023). First, we would like to clarify that, in fact, Fu et al. (2023) reported the significant effect of word embeddings for word processing in language regions, largely consistent with other studies, although specific regions varied, which might be related to the types of words being analyzed (Carota et al., 2017; Carota et al., 2024; Carota et al., 2021; Liuzzi et al., 2023). Fu et al. (2023) went further to see if such embedding representations account for neural activities specifically, beyond a set of graph-based word distances, and reported no unique effects of the embedding space.

The current study, while also starting with embedding representations, measures specifically the embedding space distance to theoretically driven anchor words, highlighting the importance of the neurobiologically related anchor words in capturing the cross-linguistic similarities. Thus, the results across studies are not in conflict.

This comment, nonetheless, prompted us to appreciate that other ways of computing distances to the anchor words, including those graph-based distances tested in Fu et al. (2023) and Mao et al. (2024), could serve as additional candidate semantic computation approaches. We have now added this point in the Discussion. Thank you for this insightful comment.

Brown, C.H. (2005). Hand and arm. In M. Haspelmath, M.S. Dryer, D. Gil, & B. Comrie

(eds.), *The world atlas of language structures* (pp. 522-525). Oxford, UK: Oxford University Press.

Carota, F., Kriegeskorte, N., Nili, H., & Pulvermüller, F. (2017). Representational similarity mapping of distributional semantics in left inferior frontal, middle temporal, and motor cortex. *Cerebral Cortex*, 27, 294-309.

Carota, F., Nili, H., Pulvermüller, F., & Kriegeskorte, N. (2021). Distinct fronto-temporal substrates of distributional and taxonomic similarity among words: Evidence from RSA of BOLD signals. *NeuroImage*, 224, 117408.

Carota, F., Nili, H., Kriegeskorte, N., & Pulvermüller, F. (2024). Experientially grounded and distributional semantic vectors uncover dissociable representations of conceptual categories. *Language, Cognition, and Neuroscience*, 39, 1020-1044.

Dellert, J., Daneyko, T., Münch, A., Ladygina, A., Buch, A., Clarius, N., et al. (2020). NorthEuraLex: A wide-coverage lexical database of Northern Eurasia. *Language Resources and Evaluation*, 54, 273-301.

Fu, Z., Wang, X., Wang, X., Yang, H., Wang, J., Wei, T., Liao, X., Liu, Z, Chen, H., & Bi, Y. (2023). Different computational relations in language are captured by distinct brain systems. *Cerebral Cortex*, 33, 997-1013.

Liuzzi, A.G., Meersmans, K., Storms, G., De Deyne, S., Dupont, P., & Vandenberghe, R. (2023). Independency of coding for affective similarities and for word co-occurrences in temporal perisylvian neocortex. *Neurobiology of Language*, 4, 257-279.

Majid, A., Bowerman, M., Kita, S., Haun, D.B.M., & Levinson, S.C. (2004). Can language restructure cognition? The case for space. *Trends in Cognitive Sciences*, 8, 108-114.

Majid, A., Jordan, F., & Dunn, M. (2015). Semantic systems in closely related languages. *Language Sciences*, 49, 1-18.

Thompson, B., Roberts, S. G., & Lupyan, G. (2020). Cultural influences on word meanings revealed through large-scale semantic alignment. *Nature Human Behaviour*, 4(10), 1029-1038.

Reviewer #5 (Remarks on code availability):

none

Reviewer #6 (Remarks to the Author):

Response: Thank you!

References:

- Artetxe, M., Labaka, G., & Agirre, E. (2018). A robust self-learning method for fully unsupervised cross-lingual mappings of word embeddings. *arXiv preprint arXiv:1805.06297*.
- Berlin, B., & Kay, P. (1991). *Basic color terms: Their universality and evolution*. Univ of California Press.
- Bi, Y. (2021). Dual coding of knowledge in the human brain. *Trends in Cognitive Sciences*, 25(10), 883-895.
- Bojanowski, P., Grave, E., Joulin, A., & Mikolov, T. (2017). Enriching word vectors with subword information. *Transactions of the association for computational linguistics*, 5, 135-146.
- Brown, C. H. (2005). Hand and arm. In M. Haspelmath, Dryer, M.S., Gil, D., & Comrie, B. (Ed.), *The world atlas of language structures* (pp. 522-525). Oxford University Press.
- Buchanan, E. M., Valentine, K. D., & Maxwell, N. P. (2019). English semantic feature production norms: An extended database of 4436 concepts. *Behavior Research Methods*, 51, 1849-1863.
- Carota, F., Kriegeskorte, N., Nili, H., & Pulvermüller, F. (2017). Representational similarity mapping of distributional semantics in left inferior frontal, middle temporal, and motor cortex. *Cerebral cortex*, 27(1), 294-309.
- Carota, F., Nili, H., Kriegeskorte, N., & Pulvermüller, F. (2024). Experientially-grounded and distributional semantic vectors uncover dissociable representations of conceptual categories. *Language, Cognition and Neuroscience*, 39(8), 1020-1044.
- Carota, F., Nili, H., Pulvermüller, F., & Kriegeskorte, N. (2021). Distinct fronto-temporal substrates of distributional and taxonomic similarity among words: evidence from RSA of BOLD signals. *NeuroImage*, 224, 117408.
- Ekman, P. (1992). An argument for basic emotions. *Cognition & Emotion*, 6(3-4), 169-200.
- Fernandino, L., Binder, J. R., Desai, R. H., Pendl, S. L., Humphries, C. J., Gross, W. L., Conant, L. L., & Seidenberg, M. S. (2016). Concept representation reflects multimodal abstraction: A framework for embodied semantics. *Cerebral cortex*, 26(5), 2018-2034.
- Fernandino, L., Tong, J.-Q., Conant, L. L., Humphries, C. J., & Binder, J. R. (2022). Decoding the information structure underlying the neural representation of concepts. *Proceedings of the National Academy of Sciences*, 119(6), e2108091119.
- Fu, Z., Wang, X., Wang, X., Yang, H., Wang, J., Wei, T., Liao, X., Liu, Z., Chen, H., & Bi, Y. (2023). Different computational relations in language are captured by distinct brain systems. *Cerebral cortex*, 33(4), 997-1013.
- Grand, G., Blank, I. A., Pereira, F., & Fedorenko, E. (2022). Semantic projection recovers rich human knowledge of multiple object features from word embeddings. *Nature Human Behaviour*, 6(7), 975-987.
- He, C., Peelen, M. V., Han, Z., Lin, N., Caramazza, A., & Bi, Y. (2013). Selectivity for large

- nonmanipulable objects in scene-selective visual cortex does not require visual experience. *NeuroImage*, 79, 1-9.
- Jackson, J. C., Lindquist, K., Drabble, R., Atkinson, Q., & Watts, J. (2023). Valence-dependent mutation in lexical evolution. *Nature Human Behaviour*, 7(2), 190-199.
- Jackson, J. C., Watts, J., Henry, T. R., List, J.-M., Forkel, R., Mucha, P. J., Greenhill, S. J., Gray, R. D., & Lindquist, K. A. (2019). Emotion semantics show both cultural variation and universal structure. *Science*, 366(6472), 1517-1522.
- Kemmerer, D. (2023). Grounded cognition entails linguistic relativity: A neglected implication of a major semantic theory. *Topics in cognitive science*, 15(4), 615-647.
- Lambon-Ralph, M. A., Jefferies, E., Patterson, K., & Rogers, T. T. (2017). The neural and computational bases of semantic cognition. *Nature Reviews Neuroscience*, 18(1), 42-55.
- Lewis, M., Cahill, A., Madnani, N., & Evans, J. (2023). Local similarity and global variability characterize the semantic space of human languages. *Proceedings of the National Academy of Sciences*, 120(51), e2300986120.
- Li, Y., Wang, M.-Y., Xu, M., Xie, W.-T., Zhang, Y.-M., Yang, X.-Y., Wang, Z.-X., Song, R., Yang, L., & Ma, J.-P. (2022). High-altitude exposure and time interval perception of Chinese migrants in Tibet. *Brain Sciences*, 12(5), 585.
- Liuzzi, A. G., Meersmans, K., Storms, G., De Deyne, S., Dupont, P., & Vandenberghe, R. (2023). Independency of coding for affective similarities and for word co-occurrences in temporal perisylvian neocortex. *Neurobiology of Language*, 4(2), 257-279.
- Majid, A., Bowerman, M., Kita, S., Haun, D. B., & Levinson, S. C. (2004). Can language restructure cognition? The case for space. *Trends in Cognitive Sciences*, 8(3), 108-114.
- Majid, A., Jordan, F., & Dunn, M. (2015). Semantic systems in closely related languages. In (Vol. 49, pp. 1-18): Elsevier.
- Majid, A., Roberts, S. G., Cilissen, L., Emmorey, K., Nicodemus, B., O'grady, L., Woll, B., LeLan, B., De Sousa, H., & Cansler, B. L. (2018). Differential coding of perception in the world's languages. *Proceedings of the National Academy of Sciences*, 115(45), 11369-11376.
- Malik-Moraleda, S., Ayyash, D., Gallée, J., Affourtit, J., Hoffmann, M., Mineroff, Z., Jouravlev, O., & Fedorenko, E. (2022). An investigation across 45 languages and 12 language families reveals a universal language network. *Nature Neuroscience*, 25(8), 1014-1019.
- Mao, S., Huebner, P. A., & Willits, J. A. (2024). Spatial versus graphical representation of distributional semantic knowledge. *Psychological review*, 131(1), 104.
- Martin, A., Haxby, J. V., Lalonde, F. M., Wiggs, C. L., & Ungerleider, L. G. (1995). Discrete cortical regions associated with knowledge of color and knowledge of action. *Science*, 270(5233), 102-105.
- McRae, K., Cree, G. S., Seidenberg, M. S., & McNorgan, C. (2005). Semantic feature production norms for a large set of living and nonliving things. *Behavior*

- Research Methods*, 37(4), 547-559.
- Mikolov, T., Le, Q. V., & Sutskever, I. (2013). Exploiting similarities among languages for machine translation. *arXiv preprint arXiv:1309.4168*.
- Miyashita, Y. (2019). Perirhinal circuits for memory processing. *Nature Reviews Neuroscience*, 20(10), 577-592.
- Passmore, S., & Jordan, F. M. (2020). No universals in the cultural evolution of kinship terminology. *Evolutionary Human Sciences*, 2, e42.
- Patterson, K., Nestor, P. J., & Rogers, T. T. (2007). Where do you know what you know? The representation of semantic knowledge in the human brain. *Nature Reviews Neuroscience*, 8(12), 976-987.
- Sigismondi, F., Xu, Y., Silvestri, M., & Bottini, R. (2024). Altered grid-like coding in early blind people. *Nature Communications*, 15(1), 3476.
- Talhelm, T., Zhang, X., Oishi, S., Shimin, C., Duan, D., Lan, X., & Kitayama, S. (2014). Large-scale psychological differences within China explained by rice versus wheat agriculture. *Science*, 344(6184), 603-608.
- Thompson, B., Roberts, S. G., & Lupyan, G. (2020). Cultural influences on word meanings revealed through large-scale semantic alignment. *Nature Human Behaviour*, 4(10), 1029-1038.
- Van de Vliert, E. (2013). Climato-economic habitats support patterns of human needs, stresses, and freedoms. *Behavioral and Brain Sciences*, 36(5), 465-480.
- Wierzbicka, A. (1992). *Semantics, culture, and cognition: Universal human concepts in culture-specific configurations*. Oxford University Press, USA.

Response to Reviewers

We sincerely thank the reviewers for their thoughtful and constructive feedback on our manuscript. We have carefully revised the manuscript accordingly. Below, we provide point-by-point responses to explain how each point was addressed and the corresponding revisions.

Reviewer #4 (Remarks to the Author):

We were very pleased to read a revised version of the article, and we think the paper is now in good shape. The authors took our comments seriously and implemented all the control analyses we asked for.

The biggest issue in the previous round was the inconsistency across analyses. The authors fixed that in the revision: the paper now uses a single approach across studies 1B/2/3 (language-level RSA with mixed-effects models that include climate, culture, geography, and linguistic history in the same model, plus random intercepts for language families). This makes the comparisons across studies clean and interpretable. Importantly, the results appear to be robust under this analysis: climate remains a reliable predictor of cross-lingual semantic alignment across studies.

Overall, we think that this is a strong revision. From our side, this looks publishable.

Response: Thank you!

Strengths

The authors implemented several changes that improved the paper, including (but not limited to):

The analytic procedures and exact model specifications are now comparable across experiments. This was the main concern with the previous version of the manuscript and now the authors have fixed it. The authors now also show lmer formulas that make the random effects structure clear.

The link between the “neurocognitive approach” and cross-lingual similarities is now clear (ll. 108-115).

Language relatedness uses a tree-based distance (lang2vec), not a 0/1 dummy (p. 23). The terminology is now more precise (rephrased “language models” to “word embeddings” for FastText; FastText is now described as word + subword embeddings; the descriptions of the different semantic models in p. 6 are clearer; the term “semantic projection” is now used correctly).

Response: Thank you for the positive comment!

Weaknesses

We disagree with the authors' interpretation of the brain results (Study 3). As we elaborated in the previous round of review, the intact vs. degraded contrast does not indicate "neural pattern similarities" (from the authors' response), but rather, where the peak responses to language are located within the region of interest—in other words, the topographical organization of the language-selective voxels in the right ATL. The paper argues that climate modulates these topographies.

This result is surprising given that meaningful cross-lingual differences in neural activations are difficult to find on top of the substantial inter-individual variability in these patterns (especially when the numbers of data points are so low for each language). Also, we cannot fathom any sensible explanation for how climate could affect the language areas' topography. We are therefore concerned that this result is spurious—similar to the finding that cultural (as opposed to climatic) factors influence the location of domain-general reasoning voxels (Figure S7). That said, these concerns do not, in our view, preclude publication. (If the authors did want to understand this effect a little better—at least to see whether the effect has to do with the peak location of the language region, or with some features of the topography surrounding the peak—they could e.g., remove the peak cluster from each participant's fROI and see if the effect holds, or conversely, remove information from anything except for the peak cluster location to see if the effect holds.)

Response: Thank you for raising this concern about the power of current sample sizes for each language and the interpretation of Study 3. We have made thorough revisions to address these issues from three angles: (1) reframing Study 3 as exploratory throughout the manuscript, (2) clarifying our conceptual interpretation of what the neural patterns reflect, and (3) conducting the robustness analyses you suggested.

(1) Reframing Study 3 as exploratory: We agree that the limited sample size per language, despite covering multiple languages, warrants a more cautious interpretation.

We have therefore reframed Study 3 as exploratory throughout the manuscript in the following locations:

Abstract (page 2): "...Exploratory neural data (86 participants, 45 languages) showed consistent climate-related activation patterns in the right anterior temporal lobe..."

Introduction (page 5): "...in Study 3, we analyzed brain activities during language comprehension, offering preliminary insights into linking cognition and biology variations more directly with the ecological variables of interest..."

Results (page 12): In summary section, we added a sentence that acknowledges the sample size issue: "...Given the limited sample size per language, these findings warrant replication with larger cohorts..."

Discussion (page 15-16): “The neural findings warrant replication with larger-samples, given the limited sample size per language in the current dataset... Future studies should employ tasks specifically designed to isolate semantic processing across diverse languages to more precisely characterize climate-semantic neural relationships.”

(2) Conceptual clarification: We appreciate the reviewer's point about terminology. To be precise, what we measured in Study 3 is the spatial pattern of neural activation magnitudes across r-ATL voxels under the “intact vs. degraded” contrast. This captures both which voxels are activated (topographic organization) and how strongly they respond (activation magnitude). We view this pattern variation as functionally meaningful because different ATL subregions show preferential engagement for different semantic dimensions (e.g., concrete objects in ventral ATL, emotional information in medial ATL, social information in dorsolateral ATL; Visser et al., 2010; Hung et al., 2020; Wang et al., 2019). Thus, variations in the distributed activation pattern—not just peak location—could reflect differential recruitment of these semantic components.

We explicitly incorporated this clarification into the Result section when first introducing the methodology of constructing neural RDMs (page 11): “These neural RDMs capture the spatial pattern of activation magnitudes across voxels, reflecting both which voxels are activated and how strongly they respond. Given that different ATL subregions show preferential engagement for different semantic dimensions⁵⁴⁻⁵⁶, variations in these distributed activation patterns may reflect differential recruitment of semantic components.”

(3) Robustness analyses: Following your suggestion, we examined whether the climate effect on neural patterns could be attributed to differences in peak activation location versus broader distributed activation patterns. Given that peak cluster locations vary across individuals, we identified each participant's peak activation cluster (centroid of the top 10% most activated voxels within r-ATL) and calculated pairwise Euclidean distances between these centroids across all language pairs. This peak distance metric captures variation in the spatial location of maximal language-selective activation. We then tested:

Does peak location vary with climate? Peak cluster distance showed no significant correlation with climate distance ($\beta = -0.08$, 95% CI [-0.22, 0.06], $p = 0.267$), indicating that climate is not associated with systematic shifts in where peak activation occurs within r-ATL.

Does the broader activation pattern matter beyond peak location? When additionally controlling for peak cluster distance, climate continued to significantly predict neural activation patterns ($\beta = 0.13$, 95% CI [0.03, 0.22], $p = 0.007$).

These results demonstrate that climate-related neural variation reflects distributed activation patterns rather than peak location displacement. Importantly, our commonality analysis revealed significant shared variance

(38.20%, 95% CI [14.75%, 48.30%]) between climate and neurocognitive semantic representation, suggesting that climate influences which semantic dimensions are preferentially recruited during language comprehension, engaging partially distinct functional subregions within ATL.

We have revised the main analysis to include peak cluster location as a control variable (page 11): “This climate effect remained significant ($\beta = 0.13$, 95% CI [0.03, 0.22], $p = 0.007$) even when controlling for differences in peak activation location (calculated as Euclidean distances between individual peak cluster centroids), indicating that the effect reflects distributed activation patterns across r-ATL voxels rather than simple spatial displacement.”

More minor issues

We disagree with the authors’ response that evaluating fastText embeddings in the native space is “challenging due to the non-isomorphic nature of embedding spaces across languages” (p. 14 of the response). This is not an issue for RSA, since it does not require direct alignment of embedding spaces. What matters is the relative structure of pairwise distances within each language’s embedding space. RSA evaluates whether the representational geometry / similarity structure is comparable, independent of absolute coordinate systems. It might be valuable (though not strictly necessary) to include an additional analysis using fastText embeddings in the native space.

Response: We thank you for this clarification and completely agree that doing RSA does not require direct alignment of embedding spaces. We would like to clarify that our “global distributional model” is precisely the native-space fastText analysis the reviewer describes. Specifically, for each language, we used the full 300-dimensional fastText embeddings in their native space to construct each concept’s representational dissimilarity vector based on pairwise cosine distances with all other 1,015 concepts. These vectors capture the relational structure of concepts within each language’s distributional semantic space — this is what we understand the reviewer refers to as “native space”.

For the results of universal comparisons, cross-language similarities of native-space distributional models are more variable than neurocognitive semantic models (shared PC1 component across languages: global-distributional = 34.16%, neurocognitive = 44.31%, page 7), indicating more language-specific information is contained in the original native space. For explaining cross-language variations, climate shows effects on both types of semantic representations: (1) climate influences the native-space distributional representations themselves when controlling neurocognitive semantic structure (i.e., variations that are not captured by neurocognitive models; Table 1 and page 9: $\beta = 0.04$, 95% CI [0.02, 0.06], $P < 7.57 \times 10^{-7}$), and (2) climate shows unique effects on neurocognitive semantic structure even when controlling for the native-

space distributional model (Table 1 and page 9: $\beta = 0.03$, 95% CI = [0.02, 0.05], $P = 5.53 \times 10^{-5}$).

We have clarified this terminology and link directly to this model with “native space” in the Results (page 6): “A prevalent approach in cross-language semantic comparisons represents concepts by their usage contexts¹². Following this approach, we represented the 1,016 concepts using their ... global patterns (i.e., anchor words were the entire native word space)” and Methods (page 17): “distributed global models, using all the other 1,015 concepts as anchors. We computed cosine similarities between each NEL concept vector and these anchors to obtain distributional semantic representations. This approach preserves the complete representational geometry within each language's native embedding space.”

In Experiment 1, the authors implement two controls: the random words model (projecting embeddings onto 100 random words) and the random dimensions model, where embeddings are projected onto 13 pseudo-dimensions. They found that the latter, with 13 dimensions, is more aligned across languages than the former. Thus, it seems that having fewer dimensions may improve the alignment. The “neurocognitive” model, which is shown to outperform other models in alignment across languages, has fewer dimensions than the alternatives. We would recommend that the authors more directly address this issue of dimensionality. A stronger demonstration of the model’s strength would involve showing that its performance cannot be explained solely by dimensionality reduction.

Response: We thank the reviewer for this important point about dimensionality. We agree this needed to be controlled for, which is why we specifically designed the random dimension model as a dimensionality-matched control. This model has the same 13 dimensions as the neurocognitive model, yet the neurocognitive model still significantly outperforms it ($P = 0.005$, Figure 2b).

We revised the description of this model to more directly in the following section: Results (page 6): “The other is the random dimension model (dimensionality-matched control), where 100 randomly selected words were grouped into 13 dimensional “anchors” using K-means clustering, and concepts were then projected onto these pseudo dimensions, with 10,000 iterations.”

Results (page 7): “Figure 2b (right panel) illustrates that the mean inter-language correlation of neurocognitive models (depicted by red vertical line) exceeded the upper ends of distributions of random word models (depicted by light gray area; $P < 0.0001$) and the dimensionality-matched random dimension models (depicted by dark gray area; $P = 0.005$), whereas the other types of theoretical models did not demonstrate this pattern. These results demonstrated that the model’s performance stems from the specific neurocognitive dimension rather than anchor word selection and dimensionality reduction procedure per se.”

Method section (page 18): “This random dimension model provides a dimensionality-matched control, demonstrating that larger cross-language alignment is not because simply having fewer dimensions.”

Suggestions

We think the article now reads well, but there is room for improvement. A suggestion for the authors to consider is to bring climate in more prominently in the introduction and in the abstract—to us, the core contribution of the paper is showing how climate modulates semantic representations across languages, but it is only mentioned at the end of the abstract (l. 33) and the introduction.

Response: We appreciate your suggestion to bring climate in more prominently and thank you for highlighting the contribution of climate results.

We have made targeted revisions to bring the introduction of the climate factor earlier: Abstract (page 2): We strengthened the presentation of climate findings in the abstract. “...with variations along this structure being significantly and uniquely explained by climate, beyond sociocultural-centered variables. Exploratory brain activity data (86 subjects, 45 languages) showed consistent climate-related activation patterns in the right anterior temporal lobe. These results present a universal, biologically constrained semantic structure that is adaptive to climate, reconciling the classical universality and relativity debate.”

Introduction (page 3): We introduced climate much earlier in the manuscript—in the third paragraph when establishing the theoretical framework. “The biological constraints of the human brain are the result of the biological evolution of *homo sapiens* and lay the foundation for universality (see similar arguments for color space⁴, emotion space¹⁴), and such a biological structure would respond to different environmental inputs (e.g., climate), resulting in phenomenal variations.”

In sum, the paper is much improved and addresses the main concerns from the first round. The results are now presented in a consistent way, and the findings on climate and semantic alignment appear to be robust. A few open questions remain, but in our view, they do not stand in the way of publication.

Response: We sincerely thank Reviewer #4 for their constructive engagement across both review rounds, which significantly strengthened our manuscript's analytical rigor, interpretive precision, and clarity.

Reviewer #4 (Remarks on code availability):

n/a

Reviewer #5 (Remarks to the Author):

The authors adequately addressed my concerns.

Response: Thank you!

Reviewer #6 (Remarks to the Author):

Response: Thank you.

Response to Reviewers

Reviewer #4 (Remarks to the Author):

We were already pleased with the previous version of the article, and we confirm the paper is in very good shape. In the previous round of review, the authors had already addressed and resolved all the substantive issues with the article. Our main point of disagreement concerned the interpretation of the neural data (Experiment 3).

This latest revision:

- Clearly states that the brain results are exploratory (throughout the paper: abstract, introduction, results, discussion).
- Includes robustness checks showing that the location of the peak cluster (centroid of top 10% of the voxels) is not what is modulated by climate, and the effects remain significant controlling for that.
- Provides a conceptual clarification on their interpretation of the r-ATL findings. (We still have questions concerning this interpretation; see below).

Again, we think this paper is a strong contribution and could be published in its current form. We have, however, one comment regarding the interpretation of the findings (in particular, two sentences added in this latest revision). We believe those considerations have big implications, and we suggest either elaborating on that or removing those considerations altogether.

At p. 11, ll. 391–395, you have added the following text to clarify your interpretation of the r-ATL findings:

(a) “These neural RDMs capture the spatial pattern of activation magnitudes across voxels, reflecting both which voxels are activated and how strongly they respond. Given that different ATL subregions show preferential engagement for different semantic dimensions, variations in these distributed activation patterns may reflect differential recruitment of semantic components.”

Analogously, in the response letter, you write:

(b) “We view this pattern variation as functionally meaningful because different ATL subregions show preferential engagement for different semantic dimensions (e.g., concrete objects in ventral ATL, emotional information in medial ATL, social information in dorsolateral ATL; Visser et al., 2010; Hung et al., 2020; Wang et al., 2019). Thus, variations in the distributed activation pattern—not just peak location—could reflect differential recruitment of these semantic components.”

The t-map being used is a contrast obtained between intact and degraded speech. It tells how strongly each voxel responds to language vs. perceptually matched controls, and not in a particular sentence versus another, but across multiple sentences (diverse in semantic content). Following the considerations listed above, we believe the (semi-implicit) interpretation of the findings is:

(1) In the r-ATL, some voxels are preferentially tuned to some semantic dimensions (say, emotional information). (We don’t know the location of those putative meaning-tuned voxels in the dataset you are using.)

(2) Climate modulates the extent to which language comprehension engages those voxels.

Our understanding is that from these points, it follows that:

(3) Climate modulates the extent to which a given language expresses certain semantic dimensions in general—not when referring to a particular concept or when processing a particular sentence, but when processing any linguistic input.

(4) Climate does not modulate the location of the voxels tuned to semantic dimensions. The location of those meaning-tuned voxels would be assumed to be relatively stable across languages and individuals, and what changes is the extent to which those voxels are recruited by language processing—which in turn, is determined by climate.

Point (3) is not a problem per se, but it is a very strong conclusion. If this is indeed the implication you intend to make, we’d recommend stating this more clearly.

Point (4), however, seems in tension with the rest of the manuscript. The paper’s main claim is that climate modulates semantic structure across languages. However, following the interpretation from the new sentences added in (a), the semantically-tuned voxels would be

assumed to be stable across individuals and languages, and what would instead vary with climate is the precise topography of language-selective voxels identified by the localizer. This inverts the logic of the paper: semantic organization becomes fixed, and the organization of the language system depends on climate.

Given this, we would recommend removing (a) altogether. As we noted before, there is currently no convincing mechanistic account of the neural finding in Study 3, and this attempt to hint at one introduces conceptual inconsistencies with other arguments laid forth in the paper.

Besides this very specific, conceptual point, we believe the paper can be published and no further review is necessary. All other issues raised in the previous round(s) have been addressed. The analyses are fully consistent across studies, the terminology is precise, the writing is clear, and the findings are interesting. This is a very nice piece of work, and we appreciate the opportunity to read it and the care in responding to our feedback.

Response: We sincerely thank you for the thorough feedback and for recognizing the strengths of our work. To address your concerns, we have removed the sentence originally added to clarify r-ATL interpretation and ensured the neural results remain presented as exploratory evidence throughout the manuscript.